civil engineering

pre-fabricated lightweight steel frame, low-energy consumption composite wall structure, anti-seismic performance, strip-shaped composite panel, houses in villages and towns

**Author for correspondence:**
Cao Wanlin
e-mail: 07814@bjut.edu.cn

# Experimental study on seismic performance of a low-energy consumption composite wall structure of a pre-fabricated lightweight steel frame

Jia Suizi[1], Cao Wanlin[2] and Liu Zibin[3]

[1]School of Engineering and Technology, China University of Geosciences (Beijing), No. 29, Xueyuan Road, Haidian District, Beijing 100083, People's Republic of China
[2]College of Architecture and Civil Engineering, Beijing University of Technology, No. 100, Pingleyuan, Chaoyang District, Beijing 100124, People's Republic of China
[3]Shandong Provincial Architectural Design Limited Liability Company, No. 2, Jingsi Road, Jinan City, Shandong Province 250001, People's Republic of China

 JS, 0000-0001-5960-2127; CW, 0000-0003-2720-4719

This study developed a low-energy consumption composite wall structure constructed with a pre-fabricated lightweight steel frame that is suitable for houses in villages and towns and evaluated its anti-seismic performance. A low-reversed cyclic-loading test was conducted on four full-scale pre-fabricated structure specimens, including a lightweight, concrete-filled steel tube (CFST) column frame specimen (abbreviated as SFCF), a lightweight CFST column frame composite wall specimen (abbreviated as SFCFW), an H-steel column frame specimen (abbreviated as HSCF) and an H-steel column frame composite wall specimen (abbreviated as HSCFW). The failure characteristics, hysteretic behaviour, strength, rigidity, ductility and energy dissipation capacity of each specimen were compared and analysed. The results demonstrated that the pre-fabricated, double L-shaped beam–column joint with a stiffener rib which was proposed in this study worked reliably and exhibited good anti-seismic performance. The yield, ultimate and frame yield loads of the specimen SFCFW were 1.72, 1.80 and 2.03 times higher than those of specimen SFCF. The yield load, ultimate load and frame yield loads of specimen HSCFW were 1.27, 1.68 and 1.82 times higher than those of specimen HSCF. This indicates that the embedded composite wall contributed significantly to the horizontal bearing capacities of the SFCF and HSCF specimens. The embedded composite wall was

divided into multiple strip-shaped composite panels during failure and achieved a stable support for the frame in the later stages of elastoplastic deformation. The horizontal strips of the tongue-and-groove connection between the strip-shaped composite panels produced reciprocating bite displacements, and ultimately improved the structure's energy dissipation capacity significantly.

## 1. Introduction

There are numerous houses in Chinese villages and towns, and 90% of them are built by the farmers themselves. The traditional structures of these houses mainly include raw-earth wall-load-bearing, brick-earth, stone-earth, wood-earth, mixed load-bearing, timber-framed load-bearing, stone-structured and brick-built houses. These structures not only require lengthy construction periods but they also have poor anti-seismic performances. Traditional structures are associated with poor wall insulation performances. In the winter, the heating of a rural house consumes significant energy, and the indoor thermal environment is uncomfortable. In the summer, most of the rural houses use cooling appliances, which also consumes large amounts of electricity.

In summary, there are numerous houses in villages and towns in China, and their existing structures are associated with poor anti-seismic performances, they require increased energy consumptions and can no longer satisfy the housing needs. Accordingly, pre-fabricated buildings exhibit a tremendous potential for the future development of the rural areas in China [1–6].

The industry of pre-fabricated structures has developed rapidly in some countries, such as the USA, Japan and Singapore [7–9].

The concept of pre-fabricated structures developed rapidly in the USA in the early 1970s. Residential buildings mainly consist of light steel and low-rise wood structures, showing diversity, modernization and other characteristics. The pre-fabricated components, reproduction techniques and the parts manufactured and developed in the USA and their applications are at the forefront of global technological progress [10–14].

Japan (1960) introduced the concept of pre-fabrication, developed a series of residential construction industrialization policies, established unified standards for modular components and addressed the contradictions among standardization, mass production and housing diversification. Japan's pre-fabricated buildings now account for a global market share of up to 50% [15–18].

Singapore is recognized as one of the best countries in the world for solving housing problems. Its residential buildings are mostly built based on building industrialization and manufacturing technologies. The housing policies and the development of pre-fabrication technologies have led to the rapid development of industrial construction. The most typical example is the introduction of unitized pre-fabricated apartment buildings comprising 15–30 storeys, which account for 80% of the market. This housing and development board (HDB) project was forcibly pre-fabricated, and the defect-free assembly rate reached 70% [19,20].

In the early days, the development of pre-fabrication was hindered in China owing to the limitations of the country's situation, construction capacity, economic conditions and the industrialization level. As a result, the development of the industrialization in the building construction in China lags behind those of the more developed countries. At present, China's industrial buildings are mainly multi-storey and high-rise residential structural systems [21].

There is no strict distinction between the studies of urban and rural buildings in developed countries. In some of these countries, only one- or two-storey industrial buildings exist in urban suburbs and villages and towns. These buildings obviously have different forms compared to traditional housing structures in rural areas in China. China's rural population is currently ageing and resource consumption is very high. The country is in a critical period regarding the promotion of energy conservation, emission reductions and green development. However, to this date, only a limited number of research studies have been conducted on the establishment of an industrialization system for rural houses that is committed to the improvement of the rural living environment and to the achievement of a sustainable development of new rural community constructions [22].

In addition, China's steel output is currently increasing while demand is falling, thus resulting in losses in the field of the steel industry. The application of lightweight steel in building rural houses not only improves their anti-seismic performances, but also alleviates the steel over-capacity dilemma. Furthermore, the use of steel in rural houses is conducive to its recycling after house demolition.

China's low-energy consumption rural housing construction exhibits significant differences with developed countries abroad in terms of resources and the respective economic levels. In our research endeavours on low-energy consumption and anti-seismic housing construction technology, we must pay special attention to economic affordability.

In summary, the research and development efforts expended on the pre-fabricated earthquake-resistant and energy-saving structures of green rural houses are highly valued by the country, and constitute a major concern for Chinese society. The development and utilization of lightweight steel and eco-friendly building materials, as well as the development of pre-fabricated low-energy consumption and earthquake-resistant rural house structures have become a major issue in the development of the beautiful rural areas in China.

At present, domestic and foreign scholars have accomplished many achievements in their studies on the combination of steel frames and masonry infill walls. Based on the continuous development of new energy-saving walls, research on the anti-seismic performance of the composite structures of energy-saving walls and steel frames is also increasing. However, research on the anti-seismic performance of low-energy consumption composite walls and lightweight steel frames is limited [23–29].

In view of the existing problems of rural house structures and the advantages of modern structures, the authors of this study propose a composite wall structure with a low-energy consumption made of a pre-fabricated lightweight steel frame. This structure is mainly applicable to green rural house structures with one to three storeys, with storey heights which are less than or equal to 4 m, and with a total height that does not exceed 10 m. These types of structures can solve the problems of poor anti-seismic performance and high-energy consumption of the current village buildings. The composite wall can also serve as a load-bearing component and concurrently provide insulation.

Considering the durability, heat preservation, fire resistance and impact resistance of the wall, the sandwich composite wall is used in the cross-sectional construction of this wall. The main types and characteristics are described next.

The first type of the composite wall is the mortar sandwich layer, which is a polystyrene granular mortar layer, while the two side layers are ordinary mortar surfaces with steel wires. The surface thicknesses are approximately equal to 20 mm, and the strength of the surface layer can attain values greater than 5 MPa. These characteristics meet the requirements of the impact resistance of the surface layer. Based on the condition of equal thickness, the weight of the wall is lighter, but the insulation effect in the sandwich layer of the polystyrene granular mortar is not as good as that of the polystyrene board.

The second type relates to the fact that the composite wall is formed by a polystyrene board sandwich layer together with fine stone concrete surfaces and steel wire meshes on both sides. Accordingly, the thickness of the surface layer is approximately 50 mm, and the strength of the surface layer can attain values greater than 20 MPa, which can meet the requirements of the impact resistance of the surface layer. However, the weight of the wall is heavier than that of the first type if the condition of equal thickness is assumed to be valid.

The third type refers to the composite wall developed in this study, whereby the middle layer is an insulation layer of graphite polystyrene board, and the two side layers are high-performance foam concrete structural layers with steel wire meshes. The surface thickness is in the range of 50−80 mm, and the surface strength can attain values greater than 5 MPa. Accordingly, these characteristics meet the impact resistance requirements of the surface layer. Additionally, the weight of the wall is equivalent to that of the first type if the condition of equal thickness, insulation effect and fire resistance of the wall are better than those of the former types.

# 2. Overview of experiments

## 2.1. Specimen design and fabrication

Four full-scale specimens were designed. The frame types were classified into a lightweight concrete-filled steel tube (CFST) column and in H-steel column frames. All the other construction dimensions, processes and materials were the same in all the designs.

### 2.1.1. Column frame design

The lightweight CFST column frame consisted of a lightweight steel tube recycled concrete column, an H-steel beam and a double L-shaped joint with a stiffener rib. The lightweight steel tube recycled

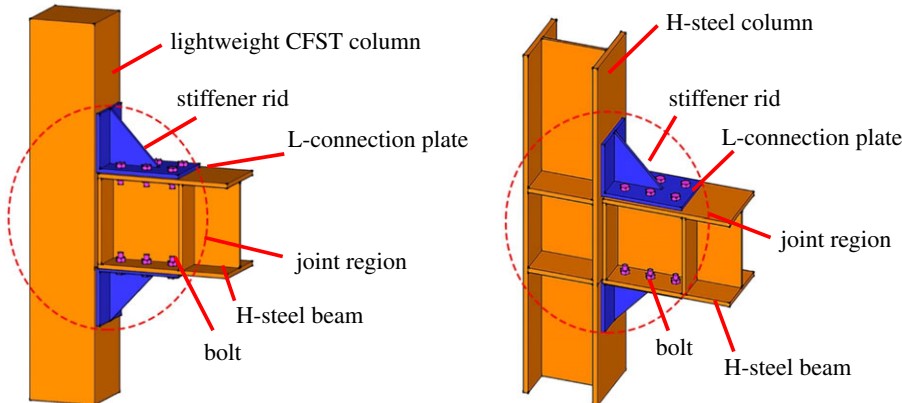

**Figure 1.** Types of column frame joint structures.

concrete column used a $150 \times 150 \times 6$ mm square steel pipe. The steel beam model was the HM model with dimensions of $194 \times 150 \times 6 \times 9$ mm. The double L-shaped joint with the stiffener rib consisted of double L-shaped components with the stiffener rib welded to a lightweight steel tube recycled concrete column. The steel connection plate of the H-steel beam and the double L-shaped joint with stiffener rib were connected with high-strength bolts (8.8 level, M12 series), and thus formed double L-shaped beam–column joints with the stiffener rib. The H-steel column frame and the lightweight CFST column frame structures were connected in the same way. The frame column model in the H-steel column frame was the HW model with dimensions of $175 \times 175 \times 7.5 \times 11$ mm. The two types of column frame joint structures are shown in figure 1.

The moment of inertia in the joint domain of the reinforced joint is significantly larger than that of the beam and column sections. Additionally, the joint is rigid, which improves the anti-lateral stiffness of the frame structure and avoids structural failures caused by the failure of the frame joint during strong earthquakes. The height of the inclined stiffener in the design should not exceed the height of the floor. Therefore, when the frame and floor were assembled together, the surface of the floor did not reveal an oblique stiffening rib. Thus, the height of the oblique stiffener rib was limited. Accordingly, the stiffener rib is only used in low-rise residence buildings at present.

When the frame and floor were assembled together, the oblique stiffener did not expose the floor surface. Therefore, the wall's filling pattern type (with the use of holes or otherwise) is irrelevant to the structure and function of the joints.

### 2.1.2. Composite wall structure design, panel and frame connection designs

The entire composite wall was constructed by splicing strip-shaped composite panels together. The strip-shaped composite panel had a length of 3540 mm, a width of 590 mm and a thickness of 240 mm. The panel consisted of three layers. The middle layer was an 80 mm thick insulation layer of a graphite polystyrene board. The two side layers comprised 80 mm thick foam concrete structural layers. The structural layer contained a $\phi3@50$ galvanized cold drawn steel mesh, and the protective layer thickness was 30 mm. One side of the foam's concrete structural layer was connected to the other side of the structural layer by a $\phi3$ crossed oblique steel wire that penetrated through the middle insulation layer. The $\phi6@200$ steel tie bars were anchored to the two sides of the panel. The exposed line length was 105 mm. The tongue-and-groove connection between the panels of the composite wall was realized by a mechanical concave–convex groove. The horizontal seams between panels were caulked with foam concrete paste. The connection joints of the composite panel are shown in figure 2. The steel tie bars on the two sides of the composite panel were welded to the lightweight frame column. A post-casting belt was attached between the lightweight frame and the composite panels, and the polystyrene foam concrete was then poured. A detailed illustration of the connection between the panel and the lightweight CFST frame is shown in figure 3.

The section size of the column was $150 \times 150$ mm, and the thickness of the panel was 240 mm. The thickness of the panel that protruded the column is chosen to be equal to 90 mm to attach the surface of the insulation layer to the column. Owing to the welding measures between the built-in tension reinforcement in the panel and the column, the out-of-plane instability of the structure under the transverse load was effectively prevented.

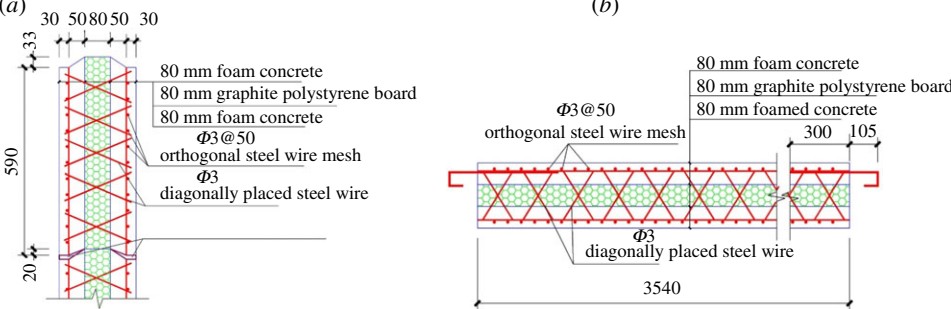

**Figure 2.** Structural design of the composite panel. (*a*) Longitudinal section and (*b*) transverse section.

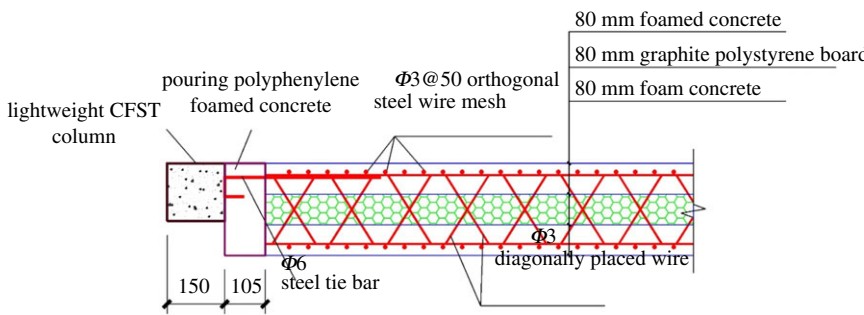

**Figure 3.** Connection between the panel and the lightweight CFST frame.

**Table 1.** Naming scheme and parameters of the prepared specimens.

| specimen no. | specifications of framework columns | specifications of framework beams | slab specifications |
|---|---|---|---|
| SFCF | lightweight recycled CFST column | H-steel beam | — |
| SFCFW | lightweight recycled CFST column | H-steel beam | composite wall |
| HSCF | H-steel column | H-steel beam | — |
| HSCFW | H-steel column | H-steel beam | composite wall |

The main parameters of the specimens are listed in table 1. The sizes and photographs of some specimens are shown in figure 4. The building heights of all the specimens were 2700 mm, and the column spacings were 3900 mm.

The assembly process of the lightweight CFST column frame composite wall structure is shown in figure 5. The assembly process of the H-steel column frame composite wall structure was the same as discussed previously.

*Frame assembly*. The frame column was fixed to the rigid foundation beam. The frame beam was then inserted into the double L-shaped joint with the use of a stiffener rib on one side. When the upper and lower flange hole positions of the beam were aligned with the L-shaped gusset hole position, high-strength bolts (M12) were tightened to complete the assembly, as shown in figure 5*a*.

*Composite wall assembly*. The strip-shaped composite panels were inserted into the frame from the side, as shown in figure 5*b*. The top convex groove of the bottom strip-shaped composite panel was inserted into the bottom concave groove of the panel above. The seam was caulked with foam concrete paste, as shown in figure 5*c*. After the strip-shaped panels had been inserted into the frame, a horizontal steel tie bar was welded to the steel tube wall of the frame column, as shown in figure 5*d*.

*Post-casting belt pouring*. The seams between the composite wall and the lightweight steel frame were filled with polystyrene foam concrete, as shown in figure 5*e*. The specimen fabrication was completed after curing and demoulding of the post-casting belt concrete, as shown in figure 5*f*.

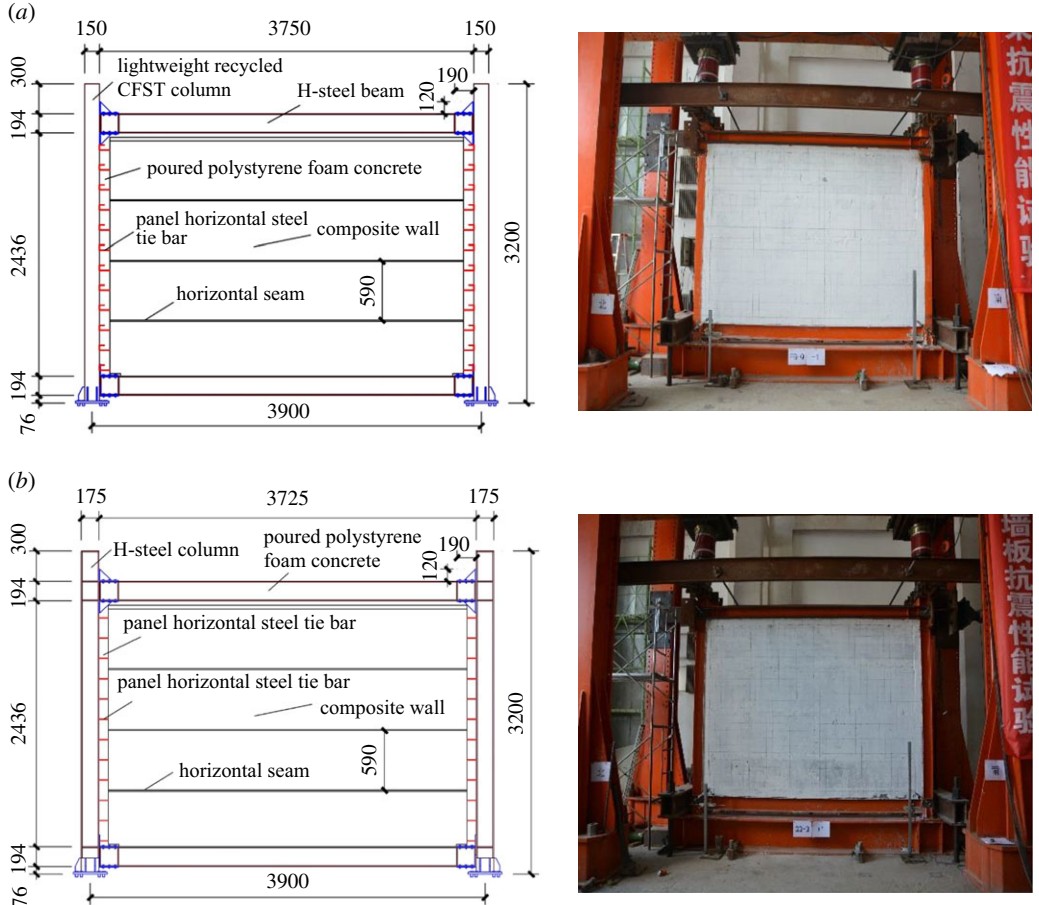

**Figure 4.** Dimensions of SFCFW and HSCFW specimens. (*a*) SFCFW and (*b*) HSCFW.

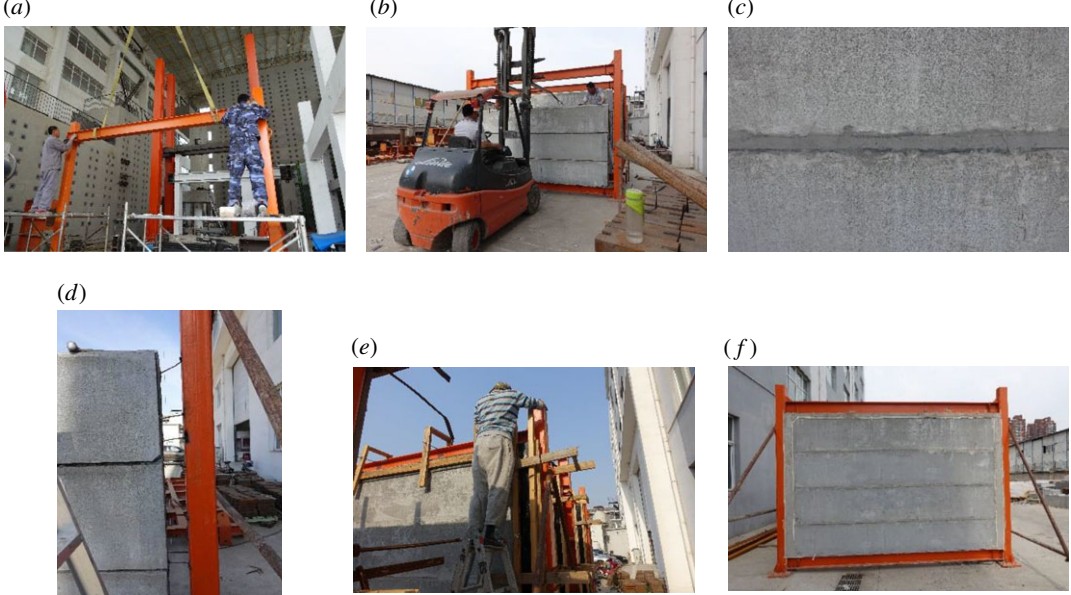

**Figure 5.** Steel frame composite wall assembly process. (*a*) Frame beam assembly, (*b*) strip-shaped composite panels hoisting, (*c*) strip-shaped panel seam caulking, (*d*) welding steel tie bar, (*e*) pouring polystyrene foam concrete and (*f*) complete fabricated specimen.

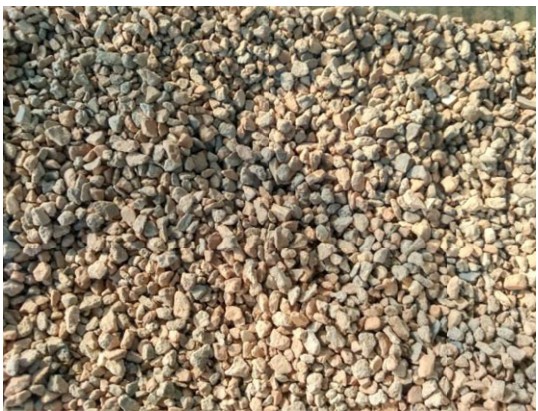

**Figure 6.** Recycled coarse aggregate (5 – 10 mm).

**Table 2.** Mix proportions of recycled concrete.

| design strength | mix proportion (kg m$^{-3}$) | | | | | | |
|---|---|---|---|---|---|---|---|
| | 42.5 cement | fly ash | mineral powder | recycled coarse aggregate | fine sand | water reducing agent | water |
| C40 | 323.0 | 70.0 | 70.0 | 804.0 | 825.0 | 4.3 | 16.5 |

**Table 3.** Physical properties of the recycled coarse aggregate.

| grain size (mm) | apparent density (kg m$^{-3}$) | water absorption (%) | crushing index (%) | void content (%) | content of elongated and flaky particles (%) |
|---|---|---|---|---|---|
| 5 – 10 | 2650 | 4.45 | 9.0 | 48.0 | 4.0 |

## 2.2. Material properties

The lightweight steel tube recycled concrete column was filled with C40 recycled concrete with a coarse aggregate replacement rate of 100%. Recycled coarse aggregate was processed and produced by the Shougang Resources Science and Technology Development Company in Beijing (figure 6). The recycled concrete mix ratio is listed in table 2. The physical properties of the coarse aggregate are listed in table 3. The fine aggregate was commercial medium sand with a moisture content of 6.02%.

According to the requirements of 'Steel and steel products—location and preparation of samples and test pieces for mechanical testing (GB/T2975-1998)' [30], samples were obtained from the corresponding locations of the test members. Three standard tensile specimens were conducted for each steel type according to the requirements of 'Metallic materials–tensile testing—part 1: method of test at room temperature (GB/T228.1-2010)' [31]. Steel with a strength grade of Q235b was used for the lightweight steel frame. Steel with a grade of HPB300 was used for the horizontal steel tie bar attached to the strip-shaped composite panels. The material property test results of the structural steel and the panel reinforcement steel are listed in table 4.

The measured cubic compressive strength foam concrete used in the strip-shaped composite panel structures was $f_{cu} = 3.8$ MPa. The measured cubic compressive strength of the post-cast polystyrene granule foam concrete was $f_{cu} = 1.9$ MPa.

The $150 \times 150 \times 150$ mm cube test block and $150 \times 150 \times 300$ mm prism test block were set aside with the same batch of concrete during pouring and were maintained using the same conditions as those used for the specimen. According to the requirements of the 'Standard for test methods for mechanical properties on ordinary concrete (GB/T50081-2002)' [32], the concrete had a measured cubic compressive strength of 54.0 MPa, an axial compressive strength of 35.5 MPa and an elastic modulus of $3.02 \times 10^4$ MPa.

The sites where the material property tests were conducted are shown in figure 7.

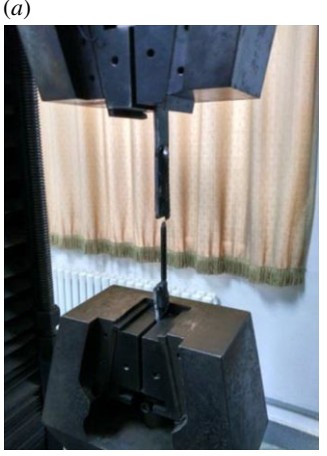

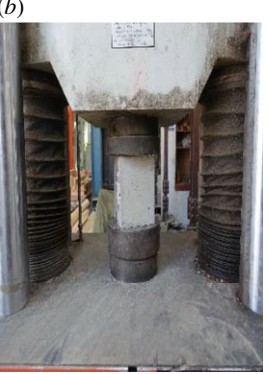

**Figure 7.** Material property tests. Material property test for (*a*) steel and (*b*) concrete.

**Table 4.** Mechanical properties of steel product.

| steel product | sampling location | diameter/ thickness $t$ (mm) | yield strength $f_y$ (MPa) | ultimate strength $f_u$ (MPa) | modulus of elasticity $E$ (GPa) | elongation $\delta$ (%) |
|---|---|---|---|---|---|---|
| steel tie bar | panel | 6 | 405.0 | 581.0 | 206.9 | 16.1 |
| galvanized cold-drawn wire | panel | 3 | 662.0 | 718.0 | 190.8 | 3.1 |
| square steel tube wall | lightweight CFST column | 6 | 373.0 | 444.3 | 218.2 | 21.5 |
| flange of steel beam | H-steel beam | 9 | 282.7 | 431.0 | 195.1 | 16.1 |
| web of steel beam | H-steel beam | 6 | 296.0 | 453.0 | 202.2 | 30.7 |
| L-connection plate | beam–column connection | 8 | 318.0 | 468.0 | 202.9 | 19.1 |
| flange of steel column | H-steel column | 11 | 270.0 | 418.0 | 202.7 | 27.5 |
| web of steel column | H-steel column | 7.5 | 360.0 | 460.0 | 204.7 | 10.0 |

## 2.3. Loading scheme and measurement point arrangements

To simulate an actual engineering load situation, a vertical load was applied on the top of each frame column, and a horizontal low-cyclic reversed load was applied along the frame beam axis. The vertical load was provided by two 250 t vertical jacks. The axial compression ratio was 0.28 (the structural system of the study was mainly suitable for low-rise (one–three storey) residential houses, the axial compression ratio of the columns was determined according to the larger axial pressure of the columns in the three-storey residential structure, and the vertical load was 499.5 kN. The vertical load was kept constant during the tests. The horizontal load was provided by a 100 t tension and compression jack. To ensure that the specimen installation process was quick and easy, the two column bases of the specimen frame columns were connected and fixed to a foundation steel beam with the use of eight high-strength bolts (grade 10.9, M20). The embedded end-effects of the column bases were simulated in this manner. A column cap was placed at the top of the specimen frame column to ensure that the vertical load could be uniformly transferred to the frame column. The specimen's out-of-plane rigidity was relatively weak. To prevent any out-of-plane instability during loading, a horizontal steel channel beam was set up to limit the specimen's lateral movement. Rigid pressure beams were configured on two of the sides of the rigid foundation beam to balance the

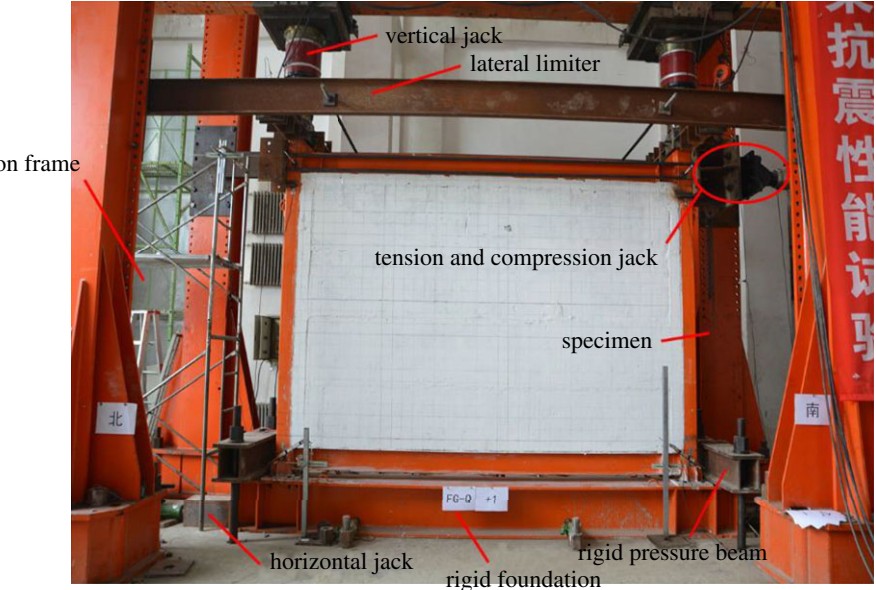

**Figure 8.** Test loading device.

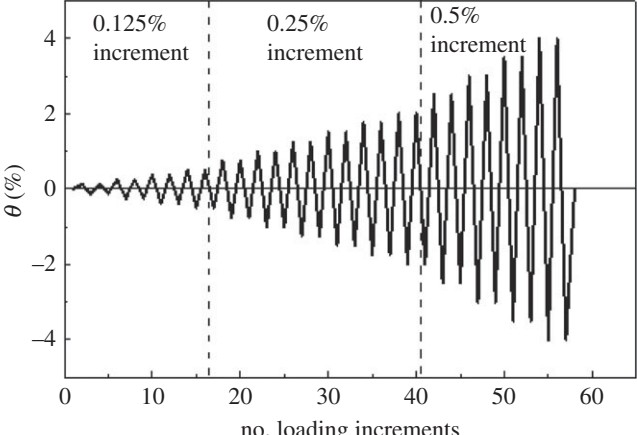

**Figure 9.** Loading scheme.

overturning moment from the horizontal load. To prevent the rigid foundation beam from sliding, horizontal jacks were placed on both ends to limit the horizontal slip.

To ensure that the wall structure did not undergo torsion or deformation outside the plane, a lateral limit device was used, which was composed of a limited steel beam which was fixed on both sides of the specimen, and a horizontal short beam which was fixed vertically on the limit steel beam end. Accordingly, the horizontal short beam was in contact with the plane of the loading column end, and could thus (i) limit the external displacement of the specimen and (ii) slip along the loading direction. This effectively prevented the torsional deformation outside the plane of the specimen's structure.

The test loading device is shown in figure 8.

The test loading was controlled based on the angular displacement $\theta$. A variable amplitude displacement loading scheme and a slow continuous loading mode were used. During loading, when the angular displacement was $\theta < 0.5\%$, the increment per stage was 0.125%. However, when $0.5\% \leq \theta \leq 2\%$, the increment per stage was 0.25%, and when $\theta > 2\%$, the increment per stage was 0.5%. Two reversed loading cycles were performed per stage. Figure 9 shows the loading scheme outcomes.

The loading experiment was terminated when one of the following conditions occurred:

(1) The bolt in the beam–column joint region of the frame experienced shear failure, or the L-shaped connection plate exhibited obvious buckling.

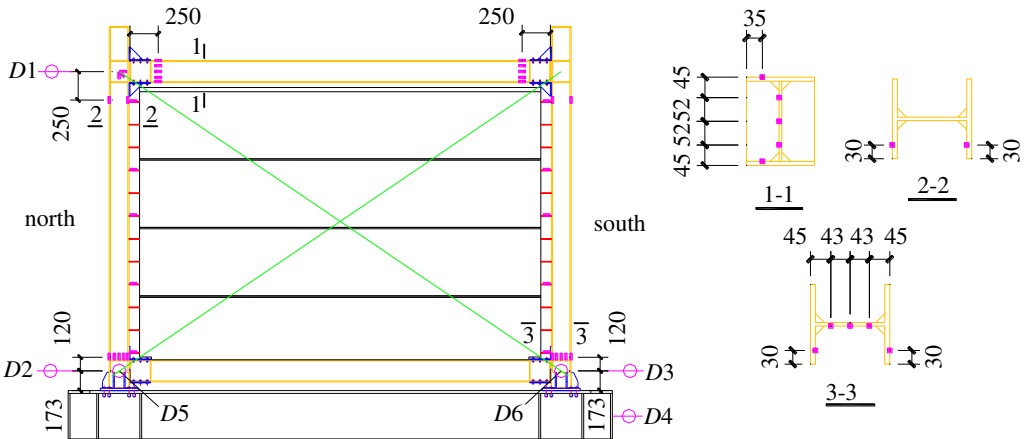

**Figure 10.** Arrangement of the measurement points.

(2) The H-steel column exhibited lateral buckling, or the steel tubes in the CFST column exhibited local buckling.

(3) The H-steel flange or web exhibited obvious buckling.

(4) The real-time horizontal load on the specimen dropped to 85% of the peak load.

Displacement gauges were placed at the height of the horizontal loading point (D1), at the heights of the foundation beam axes (D2, D3) and along the diagonal positions of the specimen (D5, D6) to measure the deformation of the walls and the frame. To eliminate the influences of the base slippage on the horizontal displacement of the specimen, a displacement gauge (D4) was placed at the height of the rigid foundation axis. To study the force transmission path of the wall and the strain response of the frame, strain gauges were placed at the end of the frame beam at the column's base, and at the wall's horizontal steel tie bar. The displacement, load and strain were measured by a static strain test system, and wall cracks were observed visually. Considering specimen HSCFW as an example, the arrangement of the strain measurement and the displacement measurement points are shown in figure 10.

# 3. Experimental phenomena and failure mechanisms

## 3.1. SFCF specimen

When the angular displacement reached 0.25%, the specimen generated a sharp sound. When it reached 0.375%, the recycled concrete in the frame column was separated from the steel pipe. When it reached 1.25%, the steel tube in the compressed frame column area generated a hollow drumming sound when it was tapped. This indicated that part of the recycled concrete in the column was crushed. When it reached 2.5%, the steel tube in the compressed frame column root exhibited slight bulging, as shown in figure 11a. The strain gauge measurement showed that the end of the H-steel beam no longer formed a plastic hinge. When it reached 4%, shear failure occurred in the high-strength bolts of the connection joint of the lightweight steel tube recycled concrete column of the H-steel beam, as shown in figure 11b. Test loading was then stopped. The specimen failure morphology is shown in figure 11c.

## 3.2. SFCFW specimen

When the angular displacement reached 0.25%, the strip-shaped composite panel seams displayed linked horizontal cracks. During reverse loading, the strip-shaped composite panels exhibited multiple parallel fine oblique cracks, as shown in figure 12a. As loading continued, the existing lengths of fine cracks extended, and a small number of new fine oblique cracks emerged. When the angular displacement reached 0.5%, slight displacement occurred among the strip-shaped composite panels. Oblique cracks occurred at the cross-sectional junction between the entire wall and the polystyrene foam concrete, as shown in figure 12b. When the angular displacement reached 0.75%, the aforementioned oblique cracks continued to widen. The largest crack was 5 mm in width and 440 mm in length. Large displacements occurred between strip panels, and the displacement distance was 5 mm. When the angular displacement reached 1.0%, the foam concrete layer at the panel seam exhibited obvious

(a)　　　　　　　　　(b)　　　　　　　　　(c)

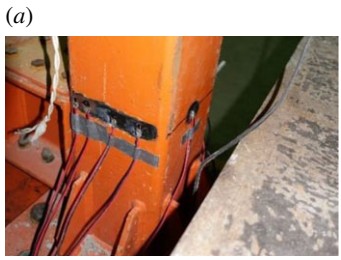 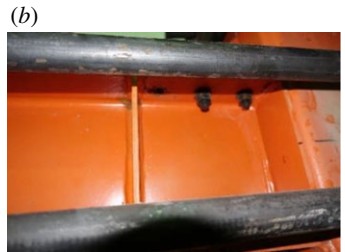 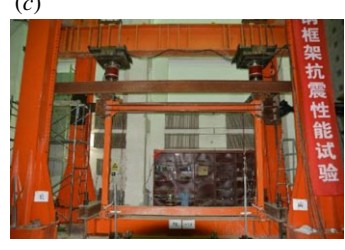

**Figure 11.** Experimental phenomena and failure of SFCF. (*a*) Steel tube bulging at the column root. (*b*) Shear failure of bolts at the beam−column joint. (*c*) Morphology of the failed structure.

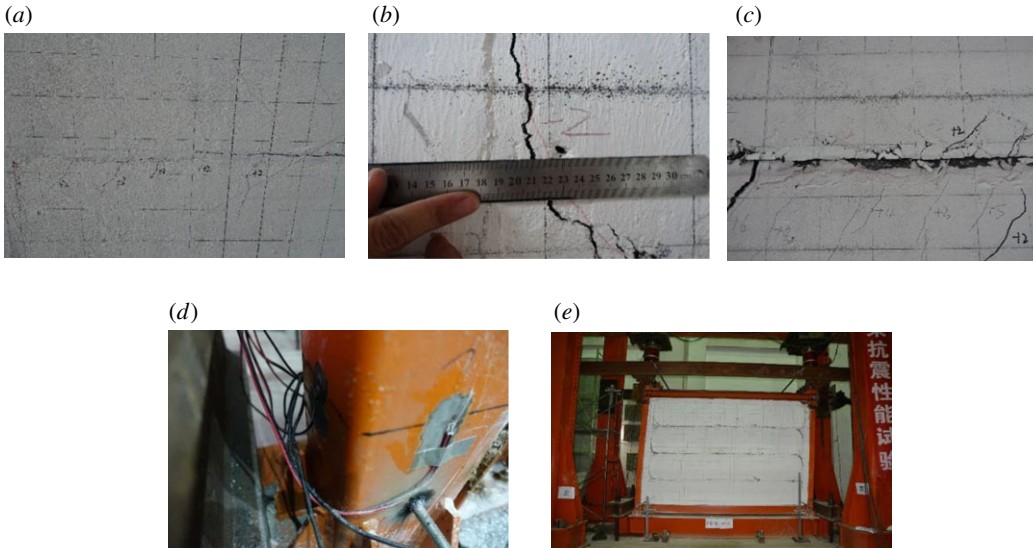

**Figure 12.** Experimental phenomena and failure of SFCFW. (*a*) Oblique cracks at the panel corner. (*b*) Widening of the oblique crack at the post-casting belt junction. (*c*) Seam foam concrete flaking. (*d*) Steel tube bulging at the column root. (*e*) Failure morphology.

flaking (figure 12*c*). When the angular displacement reached 2.0%, the steel tube at the frame column root emitted a hollow drumming sound when it was tapped. As the loading continued, the steel tube at the compressed column root showed slight bulging until significant bulging occurred (figure 12*d*). When the angular displacement reached 4%, the wall had been extensively damaged, the frame column root exhibited bulging, and it was no longer suitable for load bearing. Therefore, the test was terminated. The failure morphology of the specimen is shown in figure 12*e*.

## 3.3. HSCF and HSCFW specimens

The failure phenomenon associated with the specimen HSCF was similar to that of HSCFW of the H-steel column frame. Accordingly, HSCFW is used herein as an example to describe the failure phenomena of both tested specimens.

When the angular displacement reached 0.25%, the strip-shaped composite panel seams exhibited horizontal linked cracks with minor horizontal displacements. When the displacement angle reached 0.5%, a large number of oblique cracks occurred at the corners of the strip-shaped composite panels. The existing oblique cracks widened and extended. When the angular displacement reached 0.75%, the oblique crack at the junction between the strip-shaped composite panel and the post-casting belt became clearly visible (figure 13*a*). As the loading continued, the horizontal displacement of the strip-shaped composite panels became obvious (figure 13*b*). When the angular displacement reached 1.5%, the inner and outer flanges of the H-steel column root exhibited paint pleating (figure 13*c*). As loading continued, the pleating extended to the 450 mm range of the column root and some paint flaked off. When the angular displacement reached 3.5%, the maximum horizontal displacement distance of the strip-shaped composite panel was 30 mm. The outer flange of the column root

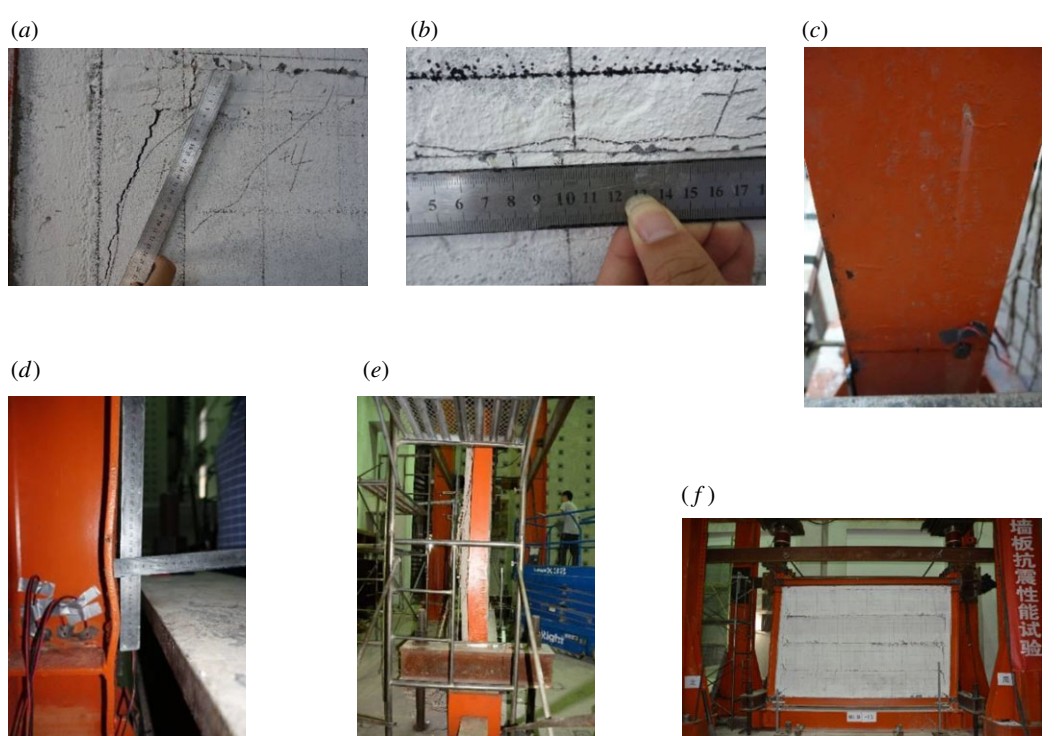

**Figure 13.** Experimental phenomena and failure of specimen HSCFW. (*a*) Oblique cracks at the corner of the wall panel. (*b*) Horizontal displacement of panel. (*c*) Paint pleating on the flange. (*d*) Flange buckling. (*e*) Column out-of-plane instability. (*f*) Failure morphology.

experienced a more severe buckling phenomenon (figure 13*d*). With continued loading, the entire column showed obvious out-of-plane instability, as shown in figure 13*e*. Similar to SFCFW, when the angular displacement of specimen HSCFW reached 3.5%, the panel was completely destroyed. Because the H-steel column exhibited significant out-of-plane instability, the structure was no longer suitable for load bearing, and the test loading was stopped. The failure morphology of the specimen is shown in figure 13*f*.

In summary, the failure processes of specimens SFCFW and HSCFW were basically the same. First, the wall was damaged, and the frame was then destroyed. The entire (embedded) composite wall first exhibited horizontal cracks at the seams of the strip-shaped composite panels, and was then displaced. When the horizontal load which was transferred to the strip-shaped composite panels became greater than the friction and bonding forces of the seam, the foam concrete at the seam cracked and produced horizontal linked cracks. Subsequently, the entire composite wall was essentially divided into multiple single strip-shaped composite panels. Part of the specimen frame column experienced bending deformation. The uncoordinated deformation of the entire embedded composite wall and the frame caused the squeezing of the single strip-shaped composite panels by the frame at the corners, which in turn led to their horizontal displacement. During the loading process, the horizontal load was transferred to the strip-shaped composite panels through the steel tie bar which was welded to the frame column. The embedded steel tie bars at the corners of the panel were subjected to reversed tension and compression. This resulted in relatively large stresses at the panel corners, and led to the generation of multiple oblique cracks. Because the deformations of the foam concrete in the strip-shaped composite panels and the post-cast polystyrene foam concrete were different, the oblique crack at the junction between the strip-shaped composite panels and the frame became wider.

The failure morphology of the frame column of specimen SFCFW was a slight bulging of the steel tube at the column root. The failure morphology of the frame column of specimen HSCFW was the compression buckling of the flange of the H-steel column and its out-of-plane instability.

In this study, the seismic performance of a lightweight CFST column frame-composite wall structure and the H-steel column frame-composite wall structure were compared. In figure 13, the steel frame column with considerable deformation outside the plane was the H-shaped steel frame column, the

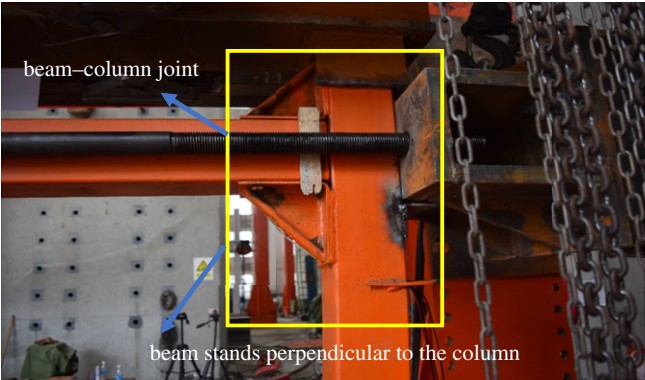

**Figure 14.** Demonstration of occurrence of damage to beam–column joints.

external deformation of the light CFST column was very small and the performance was good (figure 12). Because the composite wall adopted the easy construction of strip-shaped composite panels, the top convex groove of the bottom strip-shaped composite panel was inserted into the bottom concave groove of the panel above. The seam was caulked with foam concrete paste, the structure displacement angle was large when the seam between the strip panels staggered and consumed the seismic energy. This increased the overall deformation ability of the wall, and the self-destruction of the strip-shaped composite panel thus became lighter.

Specimens SFCF and HSCF were empty frame specimens. Before the loading stopped, the frame column root of specimen SFCF already exhibited steel tube bulging, while the H-steel column root of specimen HSCF exhibited column flange buckling under compression. The loading of specimen SFCF was stopped owing to the shear failure of the bolts at the beam–column joint, while the loading of specimen HSCF was stopped as a result of the out-of-plane instability of the H-steel column.

The double L-shaped joint with the stiffener rib of all specimens exhibited no major deformation throughout the test (figure 14). Upon the failure of the specimens, the beams and columns of the joint remained perpendicular despite the reinforced state of the joint. This would lead to an increased stiffness, which could be explained by the addition of stiffening ribs to the L-shaped joint. The design improved the strength and the stiffness of the joint areas, as the composite structure complied with the model of weak members and strong joints.

# 4. Experimental results and analyses

## 4.1. Hysteresis curve

The measured load ($F$(kN))–displacement ($\Delta$(mm)) curves of all specimens are shown in figure 15.

It can be observed in figure 15 that the hysteresis curves of specimens SFCF and HSCF both exhibit shuttle shape responses and were relatively plump. No obvious pinching phenomena were observed. This indicates that both the lightweight CFST and the H-steel column frames have good energy dissipation capacities. The hysteretic curves of both specimens could reach a maximum angular displacement of 3.5%, while the horizontal loads applied on them did not decrease significantly. This indicates that these two types of frames exhibit good deformability. The hysteretic curve of specimen SFCF was significantly different from that of SFCFW. During loading, no contraflexure points were observed in the hysteretic curve and their bearing capacities were low. Comparison of the slopes of the loading curves in the same directions indicates that the slope of the latter loading stage decreased compared to that of the previous loading stage.

During each of the loading phases of specimen SFCFW, the slope of the curve first increased at increased displacements, and a point of contraflexure then occurred before the target displacement was reached. The slope subsequently decreased at increased displacements. Comparison of the loading curves in the same direction indicates that the slope of the latter loading stage was significantly lower than that of the previous loading stage. After the reversal of the loading direction, the specimen's hysteretic curve gradually became plump, and its shape changed from an anti-S to a Z-shape.

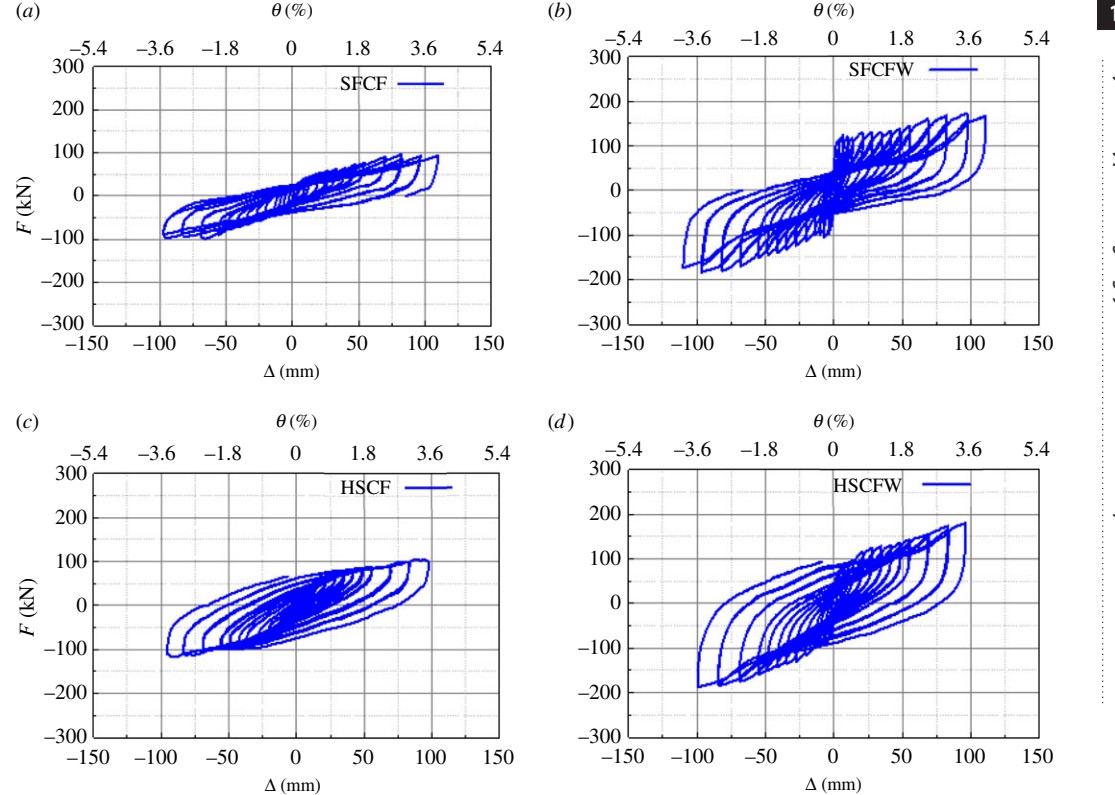

**Figure 15.** Hysteretic curves of the tested specimens. (*a*) SFCF, (*b*) SFCFW, (*c*) HSCF, (*d*) HSCFW.

The hysteretic curve of the specimen HSCFW was plumper than that of specimen HSCF and exhibited a shuttle shape. No pinching phenomenon was observed. This indicated that the specimen HSCFW had an excellent energy dissipation capacity. For the loading curves of the same stage, the bearing capacity of HSCFW was higher than that of HSCF. This indicated that the composite wall had a positive effect on the H-steel column frame.

Additionally, the authors have carried out a shaking table test on a full-scale and two-storey light CFST column frame composite wall structure, with a building height of 5400 mm, and plane size dimensions of 4400 × 4400 mm. The experimental results showed that the structure was subjected to an earthquake of 8°. The inter-storey displacement angle was approximately equal to 1/500, and there was no obvious damage to the structure. This showed that the structure had a good seismic performance. Accordingly, the structure exhibited functional recoverability at large earthquake intensities. Although the shape of the hysteretic curve was not full, the functionality can be recovered after the earthquake.

## 4.2. Skeleton curves and feature points

The load (*F*(kN))−displacement (Δ(mm)) skeleton curves of the specimens are compared in figure 16.

It can be observed in figure 16*a* that specimens SFCF and HSCF had low bearing capacity but acceptable deformability. Their skeleton curves were basically coincident. This indicated that the frame column type exerted a minor influence on the specimen's bearing capacity and deformability. The amount of steel used for the frame column of specimen HSCF was 1.49 times higher than that for specimen SFCF. However, the comparable bearing capacity and deformability of the two were equivalent. This indicated that the lightweight CFST column frame is more economical.

Figure 16*b* indicates that specimen SFCFW can be described by a four-line skeleton curve. The first segment is a rapidly rising straight line (loading began and increased until an angular displacement of 0.35% was reached). At this stage, the composite wall and the frame shared a horizontal load. The second segment was a flat line (angular displacement was between 0.35 and 0.5%). At this stage, the performance of the entire composite wall degraded rapidly and the bearing capacity no longer increased until each strip-shaped composite panel began to interact with the frame separately. In this

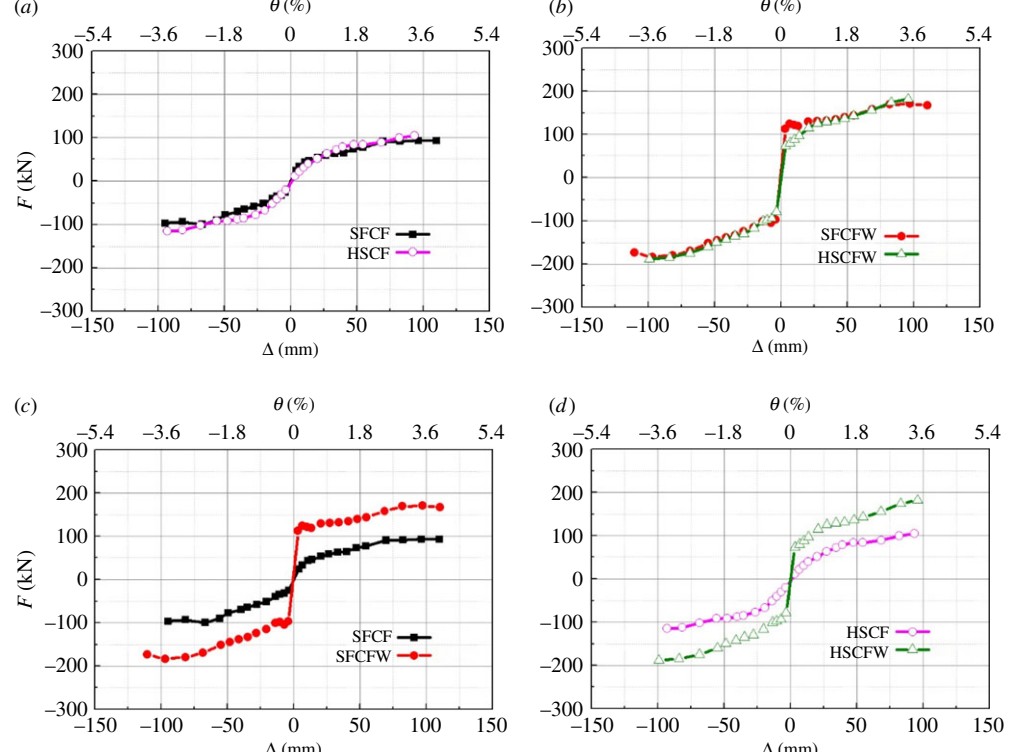

**Figure 16.** Skeleton curves of the tested specimens. (*a*) SFCF and HSCF, (*b*) SFCFW and HSCFW, (*c*) SFCF and SFCFW, (*d*) HSCF and HSCFW.

process, the staggered frictional and occlusion forces between each strip-shaped composite panel still played a role in maintaining the synergistic interaction between the entire composite wall and the frame. Overall, the specimen exhibited a plastic yield deformation with displacement increases, but yielded minor load changes. The third segment was a slowly rising straight line (the loading point changed from the angular displacement of 0.5% to the peak loading point). The frame itself entered a bearing capacity strengthening stage. The contribution of the panels to the bearing capacity could be approximated as a stable constant. The improvement of the specimen's bearing capacity depended on the strengthening of the bearing capacity of the frame. When the horizontal load on the specimen reached the peak, the bearing capacity of the frame also reached its peak. The fourth segment was a flat line (loading continued after the peak loading point). The horizontal load of the specimen did not change significantly during this stage. Because the angular displacement of the specimen was already large and because the load did not drop significantly, the test loading was stopped. The skeleton curve of specimen HSCFW could be approximately fitted to a three-line model, which consisted of a rapidly rising line segment, an oblique line and a slowly rising line segment. Compared to specimens SFCF and HSCF, the bearing capacities of specimens SFCFW and HSCFW were higher, and their deformation characteristics were equivalent to those of specimens SFCF and HSCF.

It can be observed in figure 16*c*,*d* that before the composite wall seam cracked, the composite wall significantly improved the rigidity and bearing capacity of specimens SFCFW and HSCFW. After the yielding of specimens SFCFW and HSCFW, the contribution of the strip-shaped composite panels to the structural bearing capacity was relatively stable. The differences among the bearing capacities of specimens SFCFW and HSCFW and those of SFCF and HSCF were approximately constant. After the ultimate load was reached, specimens SFCFW and HSCFW still maintained high bearing capacities that indicated that the frame of the embedded composite wall had a safety reserve.

At present, there is no uniform criterion for determining the yield and failure feature points of the H-steel and the CFST column frames of the embedded wall. In this study, the yield, ultimate and failure feature points of the specimen were calculated with the use of the universal yield bending moment method [33]. The maximum displacement during the loading process was considered as the failure displacement, and the corresponding load was considered as the failure load. The frame yield point was defined according to the yield strain of the frame column root. The method used to determine the feature points of the skeleton curve is shown in figure 17.

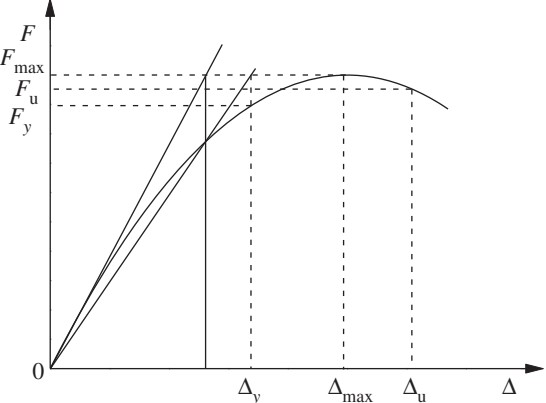

**Figure 17.** Method used for the determination of the feature points.

**Table 5.** Measured values of the feature points of the skeleton curve.

| specimen no. | yield | | ultimate | | failure | | yield of the framework | |
|---|---|---|---|---|---|---|---|---|
| | $F_y$ (kN) | $\theta_y$ (%) | $F_{max}$ (kN) | $\theta_{max}$ (%) | $F_u$ (kN) | $\theta_u$ (%) | $F_{fy}$ (kN) | $\theta_{fy}$ (%) |
| SFCF | 64.20 | 1.23 | 98.09 | 2.68 | 94.68 | 3.68 | 62.39 | 1.23 |
| SFCFW | 110.42 | 0.35 | 176.83 | 3.49 | 170.50 | 3.96 | 126.78 | 0.88 |
| HSCF | 83.70 | 1.47 | 110.07 | 3.35 | 110.07 | 3.35 | 69.88 | 0.86 |
| HSCFW | 106.31 | 0.61 | 184.75 | 3.50 | 184.75 | 3.50 | 127.50 | 1.12 |

The measured load and angular displacement of each specimen feature point are listed in table 5.

Displacement angle $\theta$ (%)–displacement at feature point ($\Delta$(mm))/distance from the embedded end of the frame column to the loading point ($h$(mm)) × 100%.

Table 5 indicates that the yield load and ultimate load of specimen SFCFW were similar to those of specimen HSCFW. This indicates that the frame column type has a minor influence on the structural bearing capacity. Both the lightweight CFST and the H-steel column frames synergistically interacted with the composite wall. The yield load, ultimate load and the frame yield load of specimen SFCFW were 1.72, 1.80 and 2.03 times higher than the corresponding values of specimen SFCF. The yield load, ultimate load and frame yield load of specimen HSCFW were 1.27, 1.68 and 1.82 times higher than the corresponding values of specimen HSCF. This indicates that the embedded composite wall provided a significant contribution to the horizontal bearing capacity of the SFCF and HSCF specimens.

## 4.3. Degradation of rigidity

This study used the secant rigidity $K_i$ to evaluate the rigidity degradation of each specimen [33]

$$K_i = \frac{|F_i^+| + |F_i^-|}{|\Delta_i^+| + |\Delta_i^-|}. \tag{4.1}$$

In the above equation, $i$ is the number of cycles, $F_i$ is the peak point load value of the $i$th cycle, $\Delta_i$ is the displacement corresponding to $F_i$ and the symbols $+$ and $-$ represent the positive and negative horizontal loadings, respectively.

The relationship curves between the measured rigidity ($K_i$) and angular displacement ($\theta$) of all specimens are shown in table 6.

It can be observed from table 6 that the rigidity values of specimens SFCF and HSCF degraded relatively slowly. The rigidity values of SFCFW and HSCFW degraded rapidly before the angular displacement of 0.5% was reached. Subsequently, the degradation slowed down. The reason for this phenomenon could be attributed to the fact that before the angular displacement of 0.5% was reached, the damage to the embedded composite wall in specimens SFCFW and HSCFW continued to increase. When the angular displacement of 0.5% was reached, the embedded composite wall was

**Table 6.** Secant stiffness values of the tested specimens.

| no. | $\theta$ (%) $K_i$ | | | | | | | | | | | | | |
| | 0.125 | 0.25 | 0.375 | 0.5 | 0.75 | 1 | 1.25 | 1.5 | 1.75 | 2 | 2.5 | 3 | 3.5 |
| --- | --- | --- | --- | --- | --- | --- | --- | --- | --- | --- | --- | --- | --- |
| SFCF | 6.23 | 4.79 | 3.63 | 3.09 | 2.55 | 2.14 | 1.87 | 1.69 | 1.56 | 1.52 | 1.40 | 1.16 | 0.99 |
| SFCFW | 29.19 | 16.81 | 10.79 | 8.13 | 5.93 | 4.58 | 3.82 | 3.30 | 2.95 | 2.68 | 2.39 | 2.13 | 1.82 |
| HSCF | 4.61 | 3.73 | 3.53 | 3.34 | 2.98 | 2.61 | 2.25 | 2.10 | 1.83 | 1.59 | 1.40 | 1.29 | 1.18 |
| HSCFW | 22.79 | 12.62 | 9.00 | 7.18 | 5.62 | 4.60 | 3.82 | 3.33 | 2.95 | 2.75 | 2.41 | 2.14 | 1.90 |

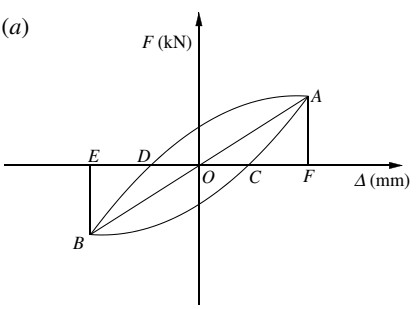

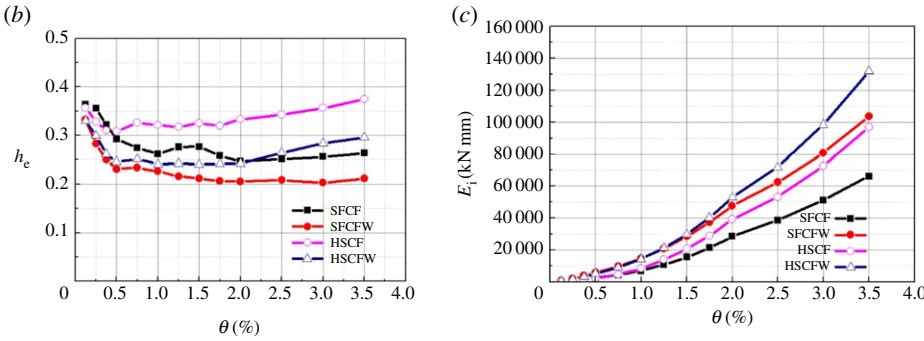

**Figure 18.** Variations of measured $h_e$ and $E$ as a function of the angular displacement of each specimen. (*a*) Force–displacement curve with hysteresis based on which $h_e$ was calculated. (*b*) The variations of the measured $h_e$ values with the angular displacement of each specimen. (*c*) The variations of the measured $E$ values with the angular displacement of each specimen.

completely destroyed and no new cracks were generated. By contrast, the rigidity degradations of specimens SFCF and HSCF were mainly caused by the buckling of the steel tube wall or the H-steel flange. Therefore, their rigidity degradation rates were relatively small. The rigidity degradation curve of specimen SFCFW is consistent with that of HSCFW, while the rigidity degradation curve of specimen SFCF is consistent with that of HSCF. This indicates that the frame column type has a minor influence on the frame rigidity degradation of the embedded composite wall.

## 4.4. Energy dissipation capacity

In all the conducted experiments, the equivalent viscous damping coefficient $h_e$ and the cumulative energy consumption $E$ were used to evaluate the energy dissipation capacities of tested specimens [34]. The cumulative energy consumption is the accumulated sum of the area of the hysteretic curve envelope of the loading cycle at each stage. The variations of the measured $h_e$ and $E$ values with the angular displacement of each specimen are shown in figure 18.

The equivalent viscous damping coefficient $h_e$ can be calculated using the following equation:

$$h_e = \frac{1}{2\pi} \frac{S_{(ACB+BDA)}}{S_{(AOF+BOE)}}, \tag{4.2}$$

where $h_e$ is the equivalent viscous damping coefficient, $S_{(ACB+BDA)}$ is the area of the hysteretic curve envelope and $S_{(AOF+BOE)}$ is the sum of the triangular areas, as shown in figure 18.

Figure 18 indicates that for the same type of frame, the $h_e$ value of the empty frame is always larger than that of the embedded composite wall frame. This is because the embedded composite wall has a restraining effect on the buckling deformation of the steel in the beam and column frames. This resulted in a hysteretic curve that is not as plump in the case of the embedded composite wall frame specimen. For the same wall construction conditions, the $h_e$ of the H-steel column frame specimen was generally larger than that of the lightweight CFST column frame specimen. This is because the recycled concrete in the steel tube has a restraining effect on the deformation of the steel. When $\theta > 2.0\%$, the $h_e$ and $E$ of specimens HSCF and HSCFW increased, but this was caused by the continuous buckling energy consumption of the H-steel column root. At this point, the column root formed a plastic hinge area. In an actual engineering project, the energy dissipation capacity of this part of the structure should not be used.

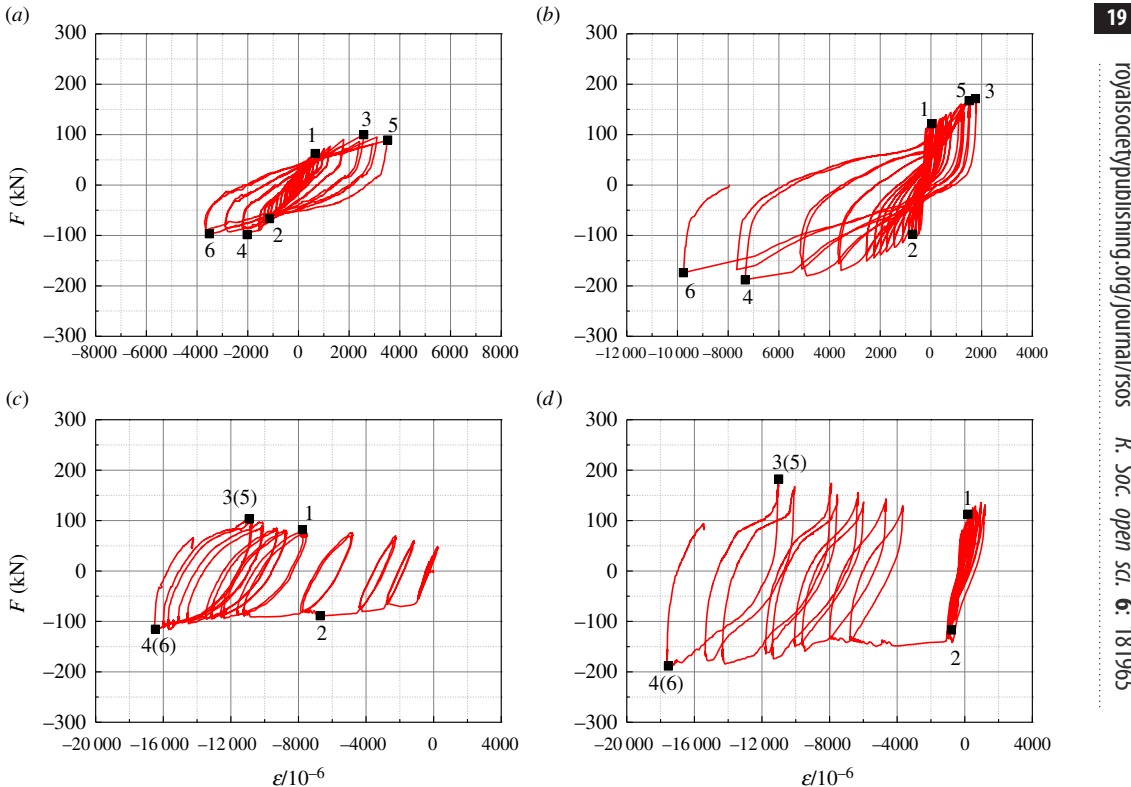

**Figure 19.** Horizontal load–strain curves of the frame column root sections of all the tested specimens. (*a*) SFCF, (*b*) SFCFW, (*c*) HSCF and (*d*) HSCFW.

**Table 7.** Energy dissipations of tested specimens.

| specimen no. | loading | $h_e$ | | $E_{max}$ (kN mm) | |
|---|---|---|---|---|---|
| | | measured value | relative values | measured value | relative values |
| SFCF | ultimate loading | 0.256 | 1.00 | 51 254.0 | 1.00 |
| SFCFW | ultimate loading | 0.212 | 0.83 | 103 517.5 | 2.02 |
| HSCF | ultimate loading | 0.375 | 1.46 | 96 720.3 | 1.89 |
| HSCFW | ultimate loading | 0.296 | 1.16 | 131 877.3 | 2.57 |

The $h_e$, $E_{max}$ and their relative values when the specimens reach the ultimate load point are shown in table 7.

Table 7 indicates that the *E*-value of specimen SFCFW was significantly larger than that of SFCF, and the *E*-value of specimen HSCFW was significantly larger than that of HSCF. This is because there were displacements between the strip-shaped composite panels. The friction energy consumption increased the total structural energy consumption. At the same time, the embedded composite wall acted synergistically with the frame and increased the energy dissipation capacity of the structure.

When the structure reached the ultimate load, the displacement angle of the specimen HSCF reaches a value of 3.5% as mentioned above, even though the values of $h_e$ and $E_{max}$ of the HSCF were, respectively, 1.46 times and 1.89 times higher than those of SFCF. Accordingly, the energy dissipation capacity of the specimen could not be used at this time.

## 4.5. Strain analysis

### 4.5.1. Horizontal load–strain relationship curve

The frame beam ends and frame column roots of all the specimens were the positions that experienced relatively large strains. Thus, the sections of the frame beam ends and the frame column roots of these

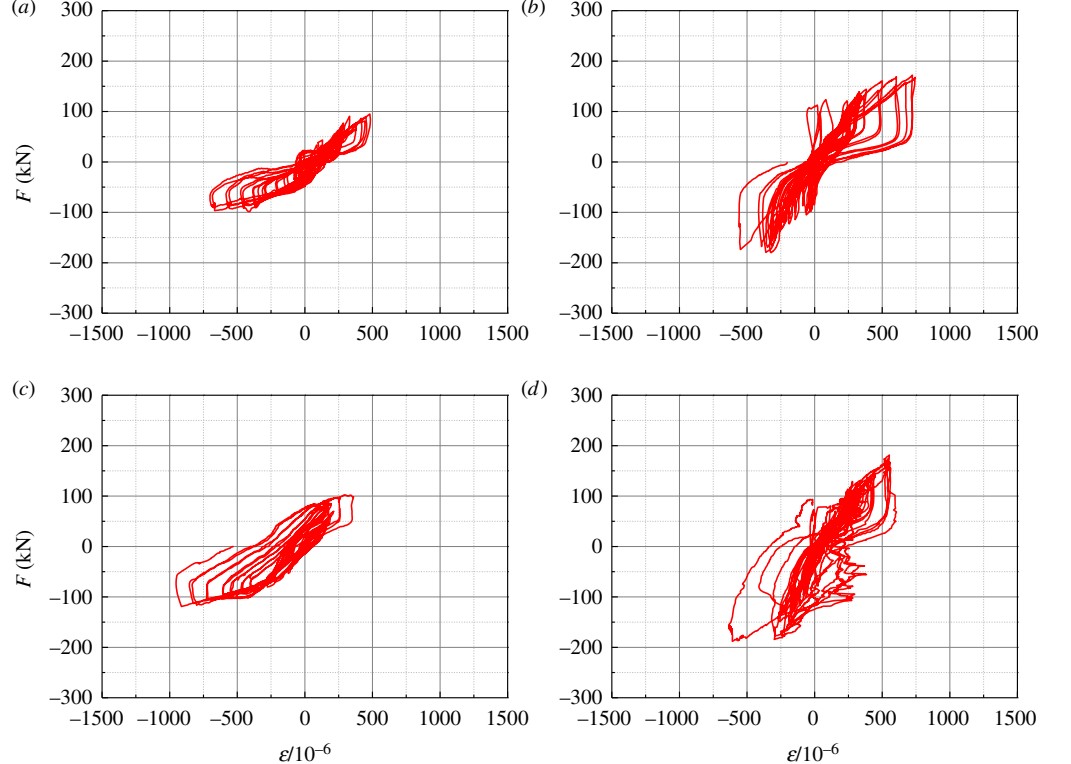

**Figure 20.** Horizontal load–strain curves of the frame beam ends of all the tested specimens. (*a*) SFCF, (*b*) SFCFW, (*c*) HSCF and (*d*) HSCFW.

specimens were used for the strain analyses. The horizontal load–strain curves of the frame column root sections of all specimens are shown in figure 19. The strain measurement point in the figure was the steel tube centre or the H-steel column flange measuring point. Points 1 and 2 in the figure are the yield load points, points 3 and 4 are the ultimate load points and points 5 and 6 are the failure load points of the specimen.

It is shown in figure 19*a*,*b* that before the specimen yielded, the steel tube basically remained in the elastic phase. Subsequently, the steel tube strain increased rapidly as a function of the horizontal load and directly entered the plastic strengthening phase. Accordingly, it did not exhibit an obvious yield point. When the SFCF reached the ultimate state, the steel tube strain was only $2000 \times 10^{-6}$ (negative compression) and $2500 \times 10^{-6}$ (positive tension). Owing to the support of the embedded composite wall, the SFCFW steel tube could be deformed considerably and fully used the advantage of the inherent strong deformability of steel.

It is shown in figure 19*c*,*d* that when specimen HSCF yielded, the strain of the H-steel column flange was greater than $6000 \times 10^{-6}$, and the column flange also yielded. However, when specimen HSCFW yielded, the strain of the H-steel flange was less than $1000 \times 10^{-6}$, and the column did not yield. This indicates that before the structural yielding, the embedded composite wall shared the horizontal load and reduced the deformation of the frame. The strain of the H-steel column root was shifted in the compression direction. This is because the H-steel column flange was continually subjected to compression buckling, so the plastic strain accumulated.

The horizontal load–strain curves of the frame beam ends are shown in figure 20. The strain measurement points in the figure are the beam-end-flange measuring points.

Figure 20 indicates that when the structure reached the failure load, none of the specimen beam-end-flange strains reached a yield strain of $1500 \times 10^{-6}$. The reason for this phenomenon is attributed to the fact that the vertical loads from the floor slab and the wall above the beam on the H-steel beam were neglected during the experiment. Only the influence of the frame column deformation on the H-steel beam was considered. Therefore, the bending moment withstood by the H-steel beam was relatively small.

### 4.5.2. Plane-section assumption

The specimens SFCF and HSCF were used to verify whether the strain change of the root sections of the lightweight steel tube recycled concrete column and H-steel column conformed to the plane-section

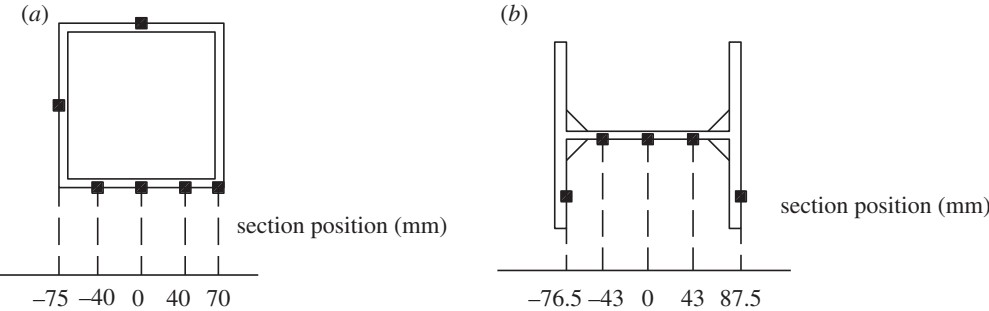

**Figure 21.** Coordinates of the position of the frame column section. (*a*) SFCF, (*b*) HSCF.

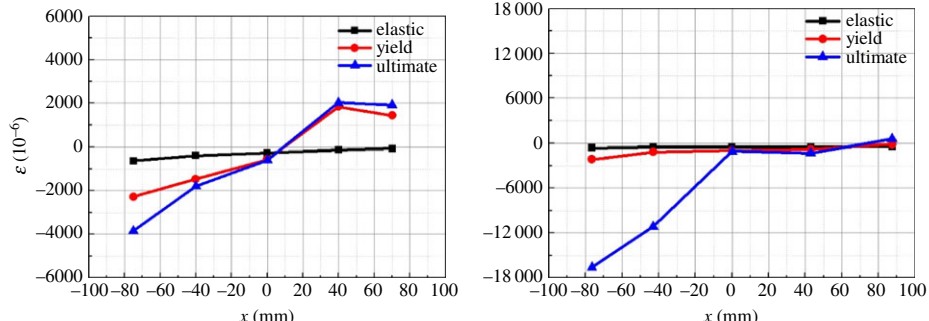

**Figure 22.** Strain distributions of the column root sections.

assumption. The coordinates of the frame column section position are defined in figure 21. The strain distributions of the column root sections are shown in figure 22.

Figure 21 shows that before specimens SFCF and HSCF yielded, the strain of the root sections of the frame columns conformed to the plane-section assumption. However, after the specimens yielded, the outermost strain of the frame column root sections increased significantly, and the section strain distributions no longer conformed to the plane-section assumption.

# 5. Conclusion

(1) The developed low-energy consumption composite wall structure with the pre-fabricated lightweight steel frame exhibited good anti-seismic performance. The proposed pre-fabricated double L-shaped joint with the stiffener rib beam–column joint worked reliably and was easy to assemble. This led to the realization of a ductile yield mechanism in the lightweight steel frame with a strong column, weak beam and a stronger joint.

(2) The developed pre-fabricated lightweight CFST column frame showed better anti-seismic performance compared to the H-steel column frame. Specifically, in the later stage of the elastoplastic deformation process, the rapid degradation of anti-seismic performance caused by the instability of light steel components was prevented.

(3) Compared to the bare lightweight CFST column frame, the yield bearing capacity of the low-energy consumption composite wall structure of the lightweight CFST column frame improved by 72%, the ultimate bearing capacity increased by 80% and the initial rigidity increased by 369%. Compared to the bare H-steel column frame, the yield bearing capacity of the low-energy consumption composite wall structure of the H-steel column frame improved by 27%, the ultimate bearing capacity increased by 68% and the initial rigidity increased by 394%. The reciprocating bite displacement of the horizontal strips of pre-fabricated strip-shaped composite panels of the tongue-and-groove connection contributed significantly to the improvement of the structural energy dissipation capacity.

Data accessibility. Data available from the Dryad Digital Repository at: https://doi.org/10.5061/dryad.cf62fp3 [35].

Authors' contributions. J.S. performed the experiment, analysed the data and wrote the manuscript; C.W. participated in the design of the experiment and data analysis; and L.Z. participated in the experiment and data analysis. All authors gave final approval for publication.

Competing interests. We declare we have no competing interests.

Funding. Financial support came from The National Natural Science Foundation of China (grant no. 51508009); The Fundamental Research Funds for the Central Universities of China (grant no. 2652017078).

Acknowledgements. This research was conducted with the support of all members of the research team for their participation in the experiment.

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
