## [Reviewer comments · Royal Society Open Science]

Review History

RSOS-181965.R0 (Original submission)

Review form: Reviewer 1

Is the manuscript scientifically sound in its present form?

Yes

Are the interpretations and conclusions justified by the results?

Yes

Is the language acceptable?

Yes

Is it clear how to access all supporting data?

Yes

Do you have any ethical concerns with this paper?

No

Have you any concerns about statistical analyses in this paper?

No

Recommendation?

Accept with minor revision (please list in comments)

Comments to the Author(s)

This manuscript presents the results of an experimental study on two prefabricated lightweight composite wall structures under cyclic lateral loading. The composite wall structures are a new type of prefabricated wall structures proposed by the authors. The test results are interesting, and topic of the study is of importance and value to the practice. I suggest acceptance after some further revision of the manuscript. Specifically, I have following comments:

1. A literature review on similar composite wall structures should be given. The advantages and rationale of the proposed composite wall structures should also be elaborated.
2. 1.2 Material Properties: more details on the recycled concrete aggregates (e.g., how and where were they obtained? How was the compressive strength of the recycled aggregates obtained?) should be provided.
3. Figure 3: Why was the size of the column significantly smaller than the thickness of the wall panel? This will cause out-of-plane instability of the structure under lateral loading.
4. 2 Experiment phenomena and failure mechanisms: the authors did not show too much information on the failure of the beam-column joints or the stiffened ribs, especially for the composite wall structures.

Review form: Reviewer 2

Is the manuscript scientifically sound in its present form?

Yes

Are the interpretations and conclusions justified by the results?

Yes

Is the language acceptable?

Yes

Is it clear how to access all supporting data?

Yes

Do you have any ethical concerns with this paper?

No

Have you any concerns about statistical analyses in this paper?

No

Recommendation?

Major revision is needed (please make suggestions in comments)

Comments to the Author(s)

Review of Manuscript ID: RSOS-181965

Title : Experimental study on seismic performance of prefabricated lightweight steel frame-low energy consumption composite wall structure

General comments: This paper presents a prefabricated lightweight steel frame-low energy consumption composite wall structure. The low reversed cyclic loading test was carried out on four full-scale specimens and evaluated their anti-seismic performance.

The failure modes, hysteretic curves, strength, ductility, and energy dissipation capacity of specimens were analyzed in detail. And the seismic performance of the structures was verified. The research conclusions can provide some technical reference for the engineering application of concrete sandwiched double steel tubes. For the engineers, this study has certain positive significance. In addition, the manuscript lacks the theoretical analysis, such as the calculation method of seismic bearing capacity of walls, etc. The manuscript can be published in the journal after the careful revision.

Besides, other evaluations and questions about this manuscript are as follow:

- 1) The abstract of the paper needs to be further condensed and it is necessary to describe the main index of seismic performance of the structures. In addition, there are some grammatical problems, please revise the abstract.
- 2) In introduction of the manuscript, the descriptive analysis on lack of the advantages of the structure proposed in the manuscript. The comparative analysis with the existing structures also needs to be strengthened appropriately. The content should be supplemented in the paper.
- 3) For ease of understanding, the graphics of the loading system need to be added.
How to determine the axial compression and whether it is reasonable?
- 4) For ease of understanding, the graphics of the loading system need to be added.
How to determine the axial compression and whether it is reasonable?
- 5) Whether the connection between steel beams and columns is rigid or semi-rigid and explain their advantages and disadvantages. How to consider the setting of doors and windows in this kind of wallboard ?
- 6) The photographs of recycled coarse aggregate and recycled concrete materials in the test need to be provided.
- 7) The numbers of specimens in this paper were limited and it needs to be supplemented by FEM simulation in the revised manuscript, to enhance the theoretical analysis.
- 8) How to ensure that the wall structure will not undergo torsion or out-of-plane deformation?
The authors are invited to supply the clarification.
- 9) The photos show that the wall damage is not serious, but the steel frame deformation is serious. How to explain? Does it meet the seismic requirements? The authors are invited to supply the clarification.
- 10) The shape of hysteretic curve of structure is not full, so it is difficult to show that the structure has good seismic performance. How to explain this question? The authors are invited to supply the clarification.
- 11) How to determine the yield displacement angle of the structure? The authors are invited to supply the clarification.
- 12) The author needs to list the main stiffness and energy dissipation indices in tabular form.
- 13) The conclusions of the paper should be combined with test-related data and some quantitative analysis conclusions need to give in this paper, which would be more convincing. The conclusions need to be streamlined and condensed again.
- 14) The text displays some mistakes in grammar, so the language expression of this manuscript must be reviewed and modified for improving the standard of English.

Decision letter (RSOS-181965.R0)

31-Jan-2019

Dear Professor Wanlin,

The editors assigned to your paper ("Experimental study on seismic performance of prefabricated lightweight steel frame-low energy consumption composite wall structure") have now received comments from reviewers. We would like you to revise your paper in accordance with the referee and Associate Editor suggestions which can be found below (not including confidential reports to the Editor). Please note this decision does not guarantee eventual acceptance.

Please submit a copy of your revised paper before 23-Feb-2019. Please note that the revision deadline will expire at 00.00am on this date. If we do not hear from you within this time then it will be assumed that the paper has been withdrawn. In exceptional circumstances, extensions may be possible if agreed with the Editorial Office in advance. We do not allow multiple rounds of revision so we urge you to make every effort to fully address all of the comments at this stage. If deemed necessary by the Editors, your manuscript will be sent back to one or more of the original reviewers for assessment. If the original reviewers are not available, we may invite new reviewers.

- Data accessibility

If you wish to submit your supporting data or code to Dryad (<http://datadryad.org/>), or modify your current submission to dryad, please use the following link:
<http://datadryad.org/submit?journalID=RSOS&manu=RSOS-181965>

- **Competing interests**

- **Authors' contributions**

- **Acknowledgements**

- **Funding statement**

on behalf of Professor R. Kerry Rowe (Subject Editor)
openscience@royalsociety.org

Associate Editor's comments:

Please ensure that you incorporate the changes requested by the reviewers, and clearly identify these in your revised manuscript. Additionally, it has been observed that your paper would benefit from the assistance of a language polishing service such as those available at <https://royalsociety.org/journals/authors/language-polishing/>. Please ensure you provide evidence of having used such a service before resubmitting. Thanks for the submission, and we await your revision.

Comments to Author:

Reviewers' Comments to Author:

Reviewer: 1

Comments to the Author(s)

This manuscript presents the results of an experimental study on two prefabricated lightweight composite wall structures under cyclic lateral loading. The composite wall structures are a new type of prefabricated wall structures proposed by the authors. The test results are interesting, and topic of the study is of importance and value to the practice. I suggest acceptance after some further revision of the manuscript. Specifically, I have following comments:

1. A literature review on similar composite wall structures should be given. The advantages and rationale of the proposed composite wall structures should also be elaborated.
2. 1.2 Material Properties: more details on the recycled concrete aggregates (e.g., how and where were they obtained? How was the compressive strength of the recycled aggregates obtained?) should be provided.
3. Figure 3: Why was the size of the column significantly smaller than the thickness of the wall panel? This will cause out-of-plane instability of the structure under lateral loading.
4. 2 Experiment phenomena and failure mechanisms: the authors did not show too much information on the failure of the beam-column joints or the stiffened ribs, especially for the composite wall structures.

Reviewer: 2

Comments to the Author(s)

Review of Manuscript ID: RSOS-181965

Title : Experimental study on seismic performance of prefabricated lightweight steel frame-low energy consumption composite wall structure

General comments: This paper presents a prefabricated lightweight steel frame-low energy consumption composite wall structure. The low reversed cyclic loading test was carried out on four full-scale specimens and evaluated their anti-seismic performance.

The failure modes, hysteretic curves, strength, ductility, and energy dissipation capacity of specimens were analyzed in detail. And the seismic performance of the structures was verified. The research conclusions can provide some technical reference for the engineering application of concrete sandwiched double steel tubes. For the engineers, this study has certain positive significance. In addition, the manuscript lacks the theoretical analysis, such as the calculation method of seismic bearing capacity of walls, etc. The manuscript can be published in the journal after the careful revision.

Besides, other evaluations and questions about this manuscript are as follow:

- 1) The abstract of the paper needs to be further condensed and it is necessary to describe the main index of seismic performance of the structures. In addition, there are some grammatical problems, please revise the abstract.
- 2) In introduction of the manuscript, the descriptive analysis on lack of the advantages of the structure proposed in the manuscript. The comparative analysis with the existing structures also needs to be strengthened appropriately. The content should be supplemented in the paper.
- 3) For ease of understanding, the graphics of the loading system need to be added.
How to determine the axial compression and whether it is reasonable?
- 4) For ease of understanding, the graphics of the loading system need to be added.
How to determine the axial compression and whether it is reasonable?

- 5) Whether the connection between steel beams and columns is rigid or semi-rigid and explain their advantages and disadvantages. How to consider the setting of doors and windows in this kind of wallboard ?
- 6) The photographs of recycled coarse aggregate and recycled concrete materials in the test need to be provided.
- 7) The numbers of specimens in this paper were limited and it needs to be supplemented by FEM simulation in the revised manuscript, to enhance the theoretical analysis.
- 8) How to ensure that the wall structure will not undergo torsion or out-of-plane deformation? The authors are invited to supply the clarification.
- 9) The photos show that the wall damage is not serious, but the steel frame deformation is serious. How to explain? Does it meet the seismic requirements? The authors are invited to supply the clarification.
- 10) The shape of hysteretic curve of structure is not full, so it is difficult to show that the structure has good seismic performance. How to explain this question? The authors are invited to supply the clarification.
- 11) How to determine the yield displacement angle of the structure? The authors are invited to supply the clarification.
- 12) The author needs to list the main stiffness and energy dissipation indices in tabular form.
- 13) The conclusions of the paper should be combined with test-related data and some quantitative analysis conclusions need to give in this paper, which would be more convincing. The conclusions need to be streamlined and condensed again.
- 14) The text displays some mistakes in grammar, so the language expression of this manuscript must be reviewed and modified for improving the standard of English.

Author's Response to Decision Letter for (RSOS-181965.R0)

See Appendix A.

RSOS-181965.R1 (Revision)

Review form: Reviewer 1

Is the manuscript scientifically sound in its present form?

Yes

Are the interpretations and conclusions justified by the results?

Yes

Is the language acceptable?

Yes

Is it clear how to access all supporting data?

Not Applicable

Do you have any ethical concerns with this paper?

No

Have you any concerns about statistical analyses in this paper?

No

Recommendation?

Accept as is

Comments to the Author(s)

The authors have addressed all my concerns. The paper can be accepted as is if all the reviewers feel the same.

Review form: Reviewer 2

Is the manuscript scientifically sound in its present form?

Yes

Are the interpretations and conclusions justified by the results?

Yes

Is the language acceptable?

Yes

Is it clear how to access all supporting data?

Yes

Do you have any ethical concerns with this paper?

No

Have you any concerns about statistical analyses in this paper?

No

Recommendation?

Accept as is

Comments to the Author(s)

The current revised manuscript can be considered for publication in the journal.

Decision letter (RSOS-181965.R1)

11-Mar-2019

Dear Professor Wanlin,

I am pleased to inform you that your manuscript entitled "Experimental study on seismic performance of a low-energy consumption composite wall structure of a pre-fabricated lightweight steel frame" is now accepted for publication in Royal Society Open Science.

on behalf of Professor R. Kerry Rowe (Subject Editor)
openscience@royalsociety.org

Associate Editor Comments to Author:
Congratulations on the acceptance of your manuscript.

Reviewer comments to Author:
Reviewer: 1

Comments to the Author(s)
The authors have addressed all my concerns. The paper can be accepted as is if all the reviewers feel the same.

Reviewer: 2

Comments to the Author(s)
The current revised manuscript can be considered for publication in the journal.

Appendix A

Response to Referees

Reviewers' Comments to Author:

Reviewer: 1

1. A literature review on similar composite wall structures should be given. The advantages and rationale of the proposed composite wall structures should also be elaborated.

Answer:

Considering the durability, heat preservation, fire resistance, and impact resistance of the wall, the sandwich composite wall is used in the cross-sectional construction of this wall. The main types and characteristics are described next.

The first type of composite wall is the mortar sandwich layer, which is a polystyrene granular mortar layer, while the two side layers are ordinary mortar surfaces with steel wires. The surface thicknesses are approximately equal to 20 mm, and the strength of the surface layer can attain values > 5 MPa. These characteristics meet the requirements of the impact resistance of the surface layer. Based on the condition of equal thickness, the weight of the wall is lighter, but the insulation effect in the sandwich layer of the polystyrene granular mortar is not as good as that of the polystyrene board.

The second type relates to the fact that the composite wall is formed by a polystyrene board sandwich layer together with fine stone concrete surfaces and steel wire meshes on both sides. Accordingly, the thickness of the surface layer is approximately 50 mm, and the strength of the surface layer can attain values > 20 MPa, which can meet the requirements of the impact resistance of the surface layer. However, the weight of the wall is heavier than that of the first type if the condition of equal thickness is assumed to be valid.

The third type refers to the composite wall developed in this study, whereby the middle layer is an insulation layer of graphite polystyrene board, and the two side layers are high-performance foam concrete structural layers with steel wire meshes. The surface thickness is in the range of 50–80 mm, and the surface strength can attain values > 5 MPa. Accordingly, these characteristics meet the impact resistance requirements of the surface layer. Additionally, the weight of the wall is equivalent to that of the first type if the condition of equal thickness, insulation effect, and fire resistance of the wall, are better than those of the former types.

2. 1.2 Material Properties: more details on the recycled concrete aggregates (e.g., how and where were they obtained? How was the compressive strength of the recycled aggregates obtained?) should be provided.

Answer:

Recycled coarse aggregate was processed and produced by the Shougang Resources Science and Technology Development Company in Beijing (Figure 6).

Figure 6 Recycled coarse aggregate (5–10 mm)

According to the requirements of 'Steel and steel products—location and preparation of samples and test pieces for mechanical testing (GB/T2975–1998)', samples were obtained from the corresponding locations of the test members. Three standard tensile specimens were conducted for each steel type according to the requirements of 'Metallic materials—tensile testing—Part 1: Method of test at room temperature (GB/T228.1–2010)'.

The 150×150×150 mm cube test block and 150×150×300 mm prism test block were set aside with the same batch of concrete during pouring and were maintained using the same conditions as those used for the specimen. According to the requirements of the 'Standard for test methods for mechanical properties on ordinary concrete (GB/T50081–2002)', the concrete had a measured cubic compressive strength of 54.0 MPa, an axial compressive strength of 35.5 MPa, and an elastic modulus of 3.02×10^4 MPa.

The sites where the material property tests were conducted are shown in Figure 7.

(a) Material property test for

steel

(b) Material property test for

concrete

Figure 7 Material property tests

3. Figure 3: Why was the size of the column significantly smaller than the thickness of the wall panel? This will cause out-of-plane instability of the structure under lateral loading.

Answer:

The section size of the column was 150 mm × 150 mm, and the thickness of the panel was 240 mm. The thickness of the panel that protruded the column is chosen to be equal to 90 mm to attach the surface of the insulation layer to the column. Owing to the welding measures between the built-in tension reinforcement in the panel and the column, the out-of-plane instability of the structure under the transverse load was effectively prevented.

4. 2 Experiment phenomena and failure mechanisms: the authors did not show too much information on the failure of the beam-column joints or the stiffened ribs, especially for the composite wall structures.

Answer:

The double L-shaped joint with the stiffener rib of all specimens exhibited no major deformation throughout the test (Figure 14). Upon the failure of the specimens, the beams and columns of the joint remained perpendicular despite the reinforced state of the joint. This would lead to an increased stiffness, which could be explained by the addition of stiffening ribs to the L-shaped joint. The design improved the strength and the stiffness of the joint areas, as the composite structure complied with the model of weak members and strong joints.

Figure 14 Demonstration of occurrence of damage to beam-column joints

Additionally, this manuscript has been reviewed and modified for improving the standard of English.

Reviewer: 2

Comments to the Author(s)

Review of Manuscript ID: RSOS-181965

Title: Experimental study on seismic performance of prefabricated lightweight steel frame-low energy consumption composite wall structure

General comments: This paper presents a prefabricated lightweight steel frame-low energy consumption composite wall structure. The low reversed cyclic loading test was carried out on four full-scale specimens and evaluated their anti-seismic performance. The failure modes, hysteretic curves, strength, ductility, and energy dissipation capacity of specimens were analyzed in detail. And the seismic performance of the structures was verified. The research conclusions can provide some technical reference for the engineering application of concrete sandwiched double steel tubes. For the engineers, this study has certain positive significance. In addition, the manuscript lacks the theoretical analysis, such as the calculation method of seismic bearing capacity of walls, etc. The manuscript can be published in the journal after the careful revision.

Besides, other evaluations and questions about this manuscript are as follow:

- 1) The abstract of the paper needs to be further condensed and it is necessary to describe the main index of seismic performance of the structures. In addition, there are some grammatical problems, please revise the abstract.

Answer:

This study developed a low-energy consumption composite wall structure constructed with a pre-fabricated lightweight steel frame that is suitable for houses in villages and towns and evaluated its anti-seismic performance. A low-reversed cyclic-loading test was conducted on four full-scale pre-fabricated structure specimens, including a lightweight,

concrete-filled steel tube (CFST) column frame specimen (abbreviated as SFCF), a lightweight CFST column frame composite wall specimen (abbreviated as SFCFW), an H-steel column frame specimen (abbreviated as HSCF), and an H-steel column frame composite wall specimen (abbreviated as HSCFW). The failure characteristics, hysteretic behaviour, strength, rigidity, ductility, and energy dissipation capacity of each specimen, were compared and analysed. The results demonstrated that the pre-fabricated, double L-shaped beam–column joint with a stiffener rib which was proposed in this study worked reliably and exhibited good anti-seismic performance. The yield, ultimate, and frame yield loads of the specimen SFCFW were 1.72, 1.80, and 2.03 times higher than those of specimen SFCF. The yield load, ultimate load, and frame yield loads of specimen HSCFW were 1.27, 1.68, and 1.82 times higher than those of specimen HSCF. This indicates that the embedded composite wall contributed significantly to the horizontal bearing capacities of the SFCF and HSCF specimens. The embedded composite wall was divided into multiple strip-shaped composite panels during failure and achieved a stable support for the frame in the later stages of elastoplastic deformation. The horizontal strips of the tongue and groove connection between the

strip-shaped composite panels produced reciprocating bite displacements, and ultimately improved the structure's energy dissipation capacity significantly.

2) In introduction of the manuscript, the descriptive analysis on lack of the advantages of the structure proposed in the manuscript. The comparative analysis with the existing structures also needs to be strengthened appropriately. The content should be supplemented in the paper.

Answer:

Considering the durability, heat preservation, fire resistance, and impact resistance of the wall, the sandwich composite wall is used in the cross-sectional construction of this wall. The main types and characteristics are described next.

The first type of composite wall is the mortar sandwich layer, which is a polystyrene granular mortar layer, while the two side layers are ordinary mortar surfaces with steel wires. The surface thicknesses are approximately equal to 20 mm, and the strength of the surface layer can attain values > 5 MPa. These characteristics meet the requirements of the impact resistance of the surface layer. Based on the condition of equal thickness, the weight of the wall is lighter, but the insulation effect in the sandwich layer of the polystyrene granular mortar is not as good as that of the polystyrene board.

The second type relates to the fact that the composite wall is formed by a polystyrene board sandwich layer together with fine stone concrete surfaces and steel wire meshes on both sides. Accordingly, the thickness of the surface layer is approximately 50 mm, and the strength of the surface layer can attain values > 20 MPa, which can meet the requirements of the

impact resistance of the surface layer. However, the weight of the wall is heavier than that of the first type if the condition of equal thickness is assumed to be valid.

The third type refers to the composite wall developed in this study, whereby the middle layer is an insulation layer of graphite polystyrene board, and the two side layers are high-performance foam concrete structural layers with steel wire meshes. The surface thickness is in the range of 50–80 mm, and the surface strength can attain values > 5 MPa. Accordingly, these characteristics meet the impact resistance requirements of the surface layer. Additionally, the weight of the wall is equivalent to that of the first type if the condition of equal thickness, insulation effect, and fire resistance of the wall, are better than those of the former types.

3) For ease of understanding, the graphics of the loading system need to be added.

How to determine the axial compression and whether it is reasonable?

Answer:

Figure 8 Loading scheme

The axial compression ratio was 0.28 (the structural system of the study was mainly suitable for low-rise (one–three story) residential houses, and the axial compression ratio of the columns was determined according to the larger axial pressure of the columns in the three-storey residential structure) and the vertical load was 499.5 kN.

4) For ease of understanding, the graphics of the loading system need to be added.

How to determine the axial compression and whether it is reasonable?

Answer:

Same as question 3

5) Whether the connection between steel beams and columns is rigid or semi-rigid and explain their advantages and disadvantages. How to consider the setting of doors and windows in this kind of wallboard ?

Answer:

The moment of inertia in the joint domain of the reinforced joint is significantly larger than that of the beam and column sections. Additionally, the joint is rigid, which improves the anti-lateral stiffness of the frame structure and avoids structural failures caused by the failure of the frame joint during strong earthquakes. The height of the inclined stiffener in the design should not exceed the height of the floor. Therefore, when the frame and floor were assembled together, the surface of the floor did not reveal an oblique stiffening rib. Thus, the height of the oblique stiffener rib was limited. Accordingly, the stiffener rib is only used in low-rise residence buildings at present.

When the frame and floor were assembled together, the oblique

stiffener did not expose the floor surface. Therefore, the wall's filling pattern type (with the use of holes or otherwise) is irrelevant to the structure and function of the joints.

6) The photographs of recycled coarse aggregate and recycled concrete materials in the test need to be provided.

Answer:

Recycled coarse aggregate was processed and produced by the Shougang Resources Science and Technology Development Company in Beijing (Figure 6).

Figure 6 Recycled coarse aggregate (5–10 mm)

According to the requirements of 'Steel and steel products–location and preparation of samples and test pieces for mechanical testing (GB/T2975–1998)', samples were obtained from the corresponding locations of the test members. Three standard tensile specimens were conducted for each steel type according to the requirements of 'Metallic materials–tensile testing–Part 1: Method of test at room temperature

(GB/T228.1–2010)'.

The 150×150×150 mm cube test block and 150×150×300 mm prism test block were set aside with the same batch of concrete during pouring and were maintained using the same conditions as those used for the specimen. According to the requirements of the 'Standard for test methods for mechanical properties on ordinary concrete (GB/T50081–2002)', the concrete had a measured cubic compressive strength of 54.0 MPa, an axial compressive strength of 35.5 MPa, and an elastic modulus of 3.02×10^4 MPa.

The sites where the material property tests were conducted are shown in Figure 7.

(a) Material property test for steel

(b) Material property test for concrete

Figure 7 Material property tests

7) The numbers of specimens in this paper were limited and it needs to be supplemented by FEM simulation in the revised manuscript, to enhance the theoretical analysis.

Answer:

It is difficult to simulate the cross-section between the tongue-groove connection. Accordingly, if the embedded composite wall is equivalent to the integral composite wall, it will not be consistent with the real composite panel technology. It is noted that the simulation error is large. The author intends to carry out more low-cycle reversed and shaking table tests on the structural system with optimised parameters, followed by the conduct of more accurate finite element simulations.

8) How to ensure that the wall structure will not undergo torsion or out-of-plane deformation? The authors are invited to supply the clarification.

Answer:

To ensure that the wall structure did not undergo torsion or deformation outside the plane, a lateral limit device was used, which was composed of a limited steel beam which was fixed on both sides of the specimen, and a horizontal short beam which was fixed vertically on the limit steel beam end. Accordingly, the horizontal short beam was in contact

with the plane of the loading column end, and could thus a) limit the external displacement of the specimen and b) slip along the loading direction. This effectively prevented the torsional deformation outside the plane of the specimen's structure.

9) The photos show that the wall damage is not serious, but the steel frame deformation is serious. How to explain? Does it meet the seismic requirements? The authors are invited to supply the clarification.

Answer:

In this study, the seismic performance of a lightweight CFST column frame-composite wall structure and the H-steel column frame-composite wall structure were compared. In the Figure 13, the steel frame column with considerable deformation outside the plane was the H-shaped steel frame column, the external deformation of the light CFST column was very small, and the performance was well (Figure 12). Because the composite wall adopted the easy construction of strip-shaped composite panels, the top convex groove of the bottom strip-shaped composite panel was inserted into the bottom concave groove of the panel above. The seam was caulked with foam concrete paste, the structure displacement angle was large when the seam between the strip panels staggered and consumed the seismic energy. This increased the overall deformation ability of the wall, and the self-destruction of the strip-shaped composite panel thus became lighter.

10) The shape of hysteretic curve of structure is not full, so it is difficult to show that the structure has good seismic performance. How to explain this question? The authors are invited to supply the clarification.

Answer:

The authors have carried out a shaking table test on a full-scale and two-story light CFST column frame composite wall structure, with a building height of 5400 mm, and plane size dimensions of 4400 mm × 4400 mm. The experimental results showed that the structure was subjected to an earthquake of 8 degree. The inter-story displacement angle was approximately equal to 1/500, and there was no obvious damage to the structure. This showed that the structure had a good seismic performance. Accordingly, the structure exhibited functional recoverability at large earthquake intensities. Although the shape of the hysteretic curve was not full, the functionality can be recovered after the earthquake.

11) How to determine the yield displacement angle of the structure? The authors are invited to supply the clarification.

Answer:

The yield, ultimate, and failure feature points of the specimen were calculated with the use of the universal yield bending moment method [30].

The method used to determine the feature points of the skeleton curve is shown in Figure 17.

Figure 17 Method used for the determination of the feature points

12) The author needs to list the main stiffness and energy dissipation indices in tabular form.

Answer:

The h_e , E_{\max} , and their relative values when the specimens reach the ultimate load point are shown in Table 7.

Table 7 Energy dissipations of tested specimens

Specimen no.	Loading	h_e		E_{\max} / (kN·mm)	
		Measured value	Relative values	Measured value	Relative values
SFCF	Ultimate		1.00		1.00
	loading	0.256		51254.0	

SFCFW	Ultimate		0.83		2.02
	loading	0.212		103517.5	
HSCF	Ultimate		1.46		1.89
	loading	0.375		96720.3	
HSCFW	Ultimate		1.16		2.57
	loading	0.296		131877.3	

Table 7 indicates that the E value of specimen SFCFW was significantly larger than that of SFCF, and the E value of specimen HSCFW was significantly larger than that of HSCF. This is because there were displacements between the strip-shaped composite panels. The friction energy consumption increased the total structural energy consumption. At the same time, the embedded composite wall acted synergistically with the frame and increased the energy dissipation capacity of the structure.

When the structure reached the ultimate load, the displacement angle of the specimen HSCF reaches a value of 3.5% as mentioned above, even though the values of h_e and E_{max} of the HSCF were respectively 1.46 times and 1.89 times higher than those of SFCF. Accordingly, the energy dissipation capacity of the specimen could not be utilised at this time.

13) The conclusions of the paper should be combined with test-related data and some quantitative analysis conclusions need to give in this paper, which would be more convincing. The conclusions need to be streamlined and condensed again.

Answer:

(1) The developed low-energy consumption composite wall structure with the pre-fabricated lightweight steel frame exhibited good anti-seismic performance. The proposed pre-fabricated double L-shaped joint with the stiffener rib beam-column joint worked reliably and was easy to assemble. This led to the realisation of a ductile yield mechanism in the lightweight steel frame with a strong column, weak beam, and a stronger joint

(2) The developed pre-fabricated lightweight CFST column frame showed better anti-seismic performance compared to the H-steel column frame. Specifically, in the later stage of the elastoplastic deformation process, the rapid degradation of anti-seismic performance caused by the instability of light steel components was prevented

(3) Compared to the bare lightweight CFST column frame, the yield bearing capacity of the low-energy consumption composite wall structure of the lightweight CFST column frame improved by 72%, the ultimate bearing capacity increased by 80%, and the initial rigidity increased by 369%. Compared to the bare H-steel column frame, the yield bearing

capacity of the low-energy consumption composite wall structure of the H-steel column frame improved by 27%, the ultimate bearing capacity increased by 68%, and the initial rigidity increased by 394%. The reciprocating bite displacement of the horizontal strips of pre-fabricated strip-shaped composite panels of the tongue and groove connection contributed significantly to the improvement of the structural energy dissipation capacity

14) The text displays some mistakes in grammar, so the language expression of this manuscript must be reviewed and modified for improving the standard of English.

Answer:

This manuscript has been reviewed and modified for improving the standard of English.

Additionally, in engineering practice, wall opening is a common phenomenon of structure, but the study of the frame with the opening wall is not mature enough. The authors plan to conduct in the future a series of opening wall tests. Based on the collected test data, the mechanical performance of the lightweight composite wall with the door

and window holes under the steel frame will be analysed and the results will be published.